# Kiaa1024L/Minar2 is essential for hearing by regulating cholesterol distribution in hair bundles

Ge Gao, Shuyu Guo, Quan Zhang, Hefei Zhang, Cuizhen Zhang, Gang Peng*

State Key Laboratory of Medical Neurobiology and MOE Frontiers Center for Brain Science, Institutes of Brain Science, Fudan University, Shanghai, China

**Abstract** Unbiased genetic screens implicated a number of uncharacterized genes in hearing loss, suggesting some biological processes required for auditory function remain unexplored. Loss of *Kiaa1024L/Minar2*, a previously understudied gene, caused deafness in mice, but how it functioned in the hearing was unclear. Here, we show that disruption of *kiaa1024L/minar2* causes hearing loss in the zebrafish. Defects in mechanotransduction, longer and thinner hair bundles, and enlarged apical lysosomes in hair cells are observed in the *kiaa1024L/minar2* mutant. In cultured cells, Kiaa1024L/Minar2 is mainly localized to lysosomes, and its overexpression recruits cholesterol and increases cholesterol labeling. Strikingly, cholesterol is highly enriched in the hair bundle membrane, and loss of *kiaa1024L/minar2* reduces cholesterol localization to the hair bundles. Lowering cholesterol levels aggravates, while increasing cholesterol levels rescues the hair cell defects in the *kiaa1024L/minar2* mutant. Therefore, cholesterol plays an essential role in hair bundles, and Kiaa1024L/Minar2 regulates cholesterol distribution and homeostasis to ensure normal hearing.

## Editor's evaluation

This is an important study linking cholesterol homeostasis to sensory hair cell function. Using the knockout approach in zebrafish, the authors provided compelling evidence that Minra2, a gene known to be associated with deafness in humans, encodes a protein that functions to regulate cholesterol homeostasis in the hair bundle of sensory hair cells.

*For correspondence:
gangpeng@fudan.edu.cn

Competing interest: The authors declare that no competing interests exist.

## Introduction

Hearing loss is one of the most common disabilities in humans (***World-Health-Organization, 2021***). Studies of genes linked to non-syndromic deafness have identified over 120 genes essential for normal hearing (***Shearer et al., 1993***; ***Van Camp and Smith, 2021***). This large collection of genes likely reflects the developmental and physiological complexity of the vertebrate auditory system. Large-scale, unbiased genetic screens in the mouse (***Bowl et al., 2017***; ***Ingham et al., 2019***) and the zebrafish (***Whitfield et al., 1996***) model systems have additionally identified multiple hearing loss genes. Interestingly, a number of these phenotypically identified hearing loss genes are previously uncharacterized and have no demonstrated functional roles, hinting that some biological processes required for normal auditory function have not been sufficiently explored (***Bowl et al., 2017***; ***Ingham et al., 2019***).

The auditory hair cells are the sensory receptor cells that convert acoustic and mechanical stimuli into electrical signals initiating hearing (***Fettiplace, 2017***; ***Hudspeth, 1989***; ***Ó Maoiléidigh and Ricci, 2019***). The hair bundle, a specialized organelle situated at the apical surface of each hair cell, responds to mechanical displacement in a direction-dependent manner (***Flock, 1964***; ***Tilney et al.,***

**eLife digest** Cholesterol is present in every cell of the body. While it is best known for its role in heart health, it also plays a major role in hearing, with changes in cholesterol levels negatively affecting this sense. To convert sound waves into electrical brain signals, specialised ear cells rely on hair-like structures which can move with vibrations; cholesterol is present within these hair 'bundles', but its exact role remains unknown.

Genetic studies have identified over 120 genes essential for normal hearing. Animal data suggest there may be many more – including, potentially, some which control cholesterol. For instance, in mice, loss of the *Minar2* gene causes profound deafness. Yet exactly which role the protein that *Minar2* codes for plays in the ear remains unknown. This is in part because that protein does not resemble any other related proteins, making it difficult to infer its function.

To find out more, Gao et al. investigated loss of *minar2* in zebrafish, showing that deleting the gene induced deafness in the animals. Without *minar2*, the hair bundles in ear cells were longer, thinner, and less able to sense vibrations: cholesterol could not move into these structures, causing them to dysfunction. Exposing the animals to drugs that lower or raise cholesterol levels respectively worsened or improved their hearing abilities.

A recent study revealed that mutations in *MINAR2* also cause deafness in humans. The findings by Gao et al. highlight the need for further research which explores the role of cholesterol and *MINAR2* in hair bundle function, as this may potentially uncover cholesterol-based treatments for hearing problems.

*1992*). The hair bundle is essential for mechanoelectrical transduction (MET), and defects in the hair bundle can cause hearing loss (*Belyantseva et al., 2009*; *Blanco-Sánchez et al., 2014*; *Kozlov et al., 2007*; *Noben-Trauth et al., 2003*; *Perrin et al., 2013*). Several proteins, such as adhesion molecules, actin-bundling proteins, and disease-associated proteins are required for the morphogenesis and physiological regulation of the hair bundle (*Barr-Gillespie, 2015*; *Blanco-Sánchez et al., 2017*; *McGrath et al., 2017*; *Tilney et al., 1980*). In addition, specific lipid molecules may play a role in the hair bundle. For instance, phosphatidylinositol-4,5-bisphosphate (PIP2) is localized to hair bundles, and it binds to the MET channel component TMIE (*Cunningham et al., 2020*) and regulates the rates of fast and slow adaptation (*Effertz et al., 2017*; *Hirono et al., 2004*).

Cholesterol is an important component of eukaryotic cell membranes, and it controls membrane stiffness, tension, fluidity, and other membrane properties (*Maxfield and van Meer, 2010*; *Subczynski et al., 2017*). Cholesterol also plays a regulatory function by interacting with membrane proteins (*Harris, 2010*). Previous studies showed that abnormally high or low cholesterol levels are detrimental to hearing (*Corwin and Warchol, 1991*; *Crumling et al., 2012*; *Ding et al., 2020*; *Guo et al., 2005*; *King et al., 2014a*; *King et al., 2014b*; *Morizono and Paparella, 1978*; *Sikora et al., 1986*; *Takahashi et al., 2016*; *Thoenes et al., 2015*; *Xing et al., 2015*; *Yao et al., 2019*). Early studies also showed that cholesterol is not uniformly distributed in the hair cell membranes. The intensities of cholesterol labeling the hair cells were higher in the apical membranes when compared to those of the lateral membranes (*Nguyen and Brownell, 1998*; *Takahashi et al., 2016*), and freeze-fracture images of hair cells suggested that the stereocilia membrane was densely covered with cholesterol (*Forge et al., 1988*). Nevertheless, the distribution of cholesterol in hair cells in vivo, and cholesterol's functional role in hair cells are not well characterized.

*Kiaa1024L/Minar2*, a previously understudied gene, was identified in a hearing loss screen in mouse knockout strains using the auditory brainstem response test. The homozygous *Kiaa1024L/Minar2* knockout mice aged 14 weeks old had severely raised ABR thresholds at all frequencies tested (*Bowl et al., 2017*; *Ingham et al., 2019*), but how *Kiaa1024L/Minar2* functioned in the hearing was unclear. Here we show *kiaa1024L/minar2* is expressed in the zebrafish mechanosensory hair cells, and disruption of *kiaa1024L/minar2* causes hearing loss in the zebrafish larvae. We next show that GFP or FLAG-tagged Kiaa1024L/Minar2 protein is distributed in the stereocilia and the apical endo-membranes. Defects in mechanotransduction, longer and thinner hair bundles, and enlarged apical lysosomes are observed in hair cells in *kiaa1024L/minar2* mutant. In vitro studies in cultured cells show that the Kiaa1024L/Minar2 protein is mainly localized to lysosomes, and overexpression of Kiaa1024L/

Minar2 recruits cholesterol and results in increased intracellular cholesterol levels. Strikingly, we show cholesterol is highly enriched in the hair bundle membranes, and loss of *kiaa1024L/minar2* reduces cholesterol distribution in the hair bundles. Drug treatment that lowers cholesterol levels aggravates, whereas treatment that raises cholesterol levels rescues hair cell defects and hearing in the mutant *kiaa1024L/minar2* larvae. Together, our results indicate cholesterol plays an essential role in the hair bundles, and Kiaa1024L/Minar2 regulates cholesterol distribution and homeostasis in auditory hair cells to ensure normal hearing.

## Results

### *kiaa1024L/minar2* is required for normal hearing in the zebrafish

*Kiaa1024L/minar2* gene orthologs are found in vertebrate species only (*Figure 1—figure supplement 1A*). It belongs to the UPF0258 gene family, which also includes *kiaa1024/minar1/ubtor* (*Ho et al., 2018*; *Zhang et al., 2018*). Based on genome annotations and BLAST search results, there are two *kiaa1024/minar1/ubtor* gene orthologs (named *ubtora* and *ubtorb* in *Zhang et al., 2018*), and a single *kiaa1024l/minar2* gene ortholog in the zebrafish genome. To conform to current human gene nomenclature, *kiaa1024L/minar2* gene orthologs are referred to as *minar2* hereafter.

We surveyed available sequencing data (*Barta et al., 2018*; *Elkon et al., 2015*; *Erickson and Nicolson, 2015*; *Liu et al., 2018*) and found the transcripts of *minar2* orthologs were highly enriched in the auditory hair cells of the mouse and the zebrafish (*Figure 1—figure supplement 1B*). In a human inner ear organoid model (*Steinhart et al., 2022*), *MINAR2* is specifically expressed in differentiated hair cells, similar to known differentiated hair cell markers (*Figure 1—figure supplement 1B*). Consistent with these sequencing-based data, in situ hybridization results confirmed *minar2* was specifically expressed by the hair cells of the inner ears and the lateral line neuromasts in the developing zebrafish (5 dpf, days post fertilization, *Figure 1A* and *Figure 1—figure supplement 1A*).

To study *minar2* function in the zebrafish, we generated *minar2* mutant alleles by CRISPR/Cas9-mediated mutagenesis. The mutation in the *minar2*[fs139] allele was a 5 bp deletion in exon 1, and in the *minar2*[fs140] allele was a 5 bp insertion in exon 1 (*Figure 1—figure supplement 1C*). Both mutations were frameshift mutations and led to premature termination of protein translation. The translatable protein sequence in either mutant was very short and lacked the transmembrane helix located at the carboxyl terminus of the protein (*Figure 1—figure supplement 1C*). The mutant Minar2[fs139] protein was predicted to translate to the 25th amino acid, then adds 28 code-shifted residues before the reading frame stopped. The mutant Minar2[fs140] protein was predicted to translate to the 26th amino acid, then adds 55 code-shifted residues. Thus, both mutants were expected to be loss of function alleles. In addition, quantitative PCR analyses showed the expression levels of *minar2* in the two mutants were markedly down-regulated in the developing zebrafish, most likely because the premature translational terminations activated the nonsense-mediated mRNA decays (*Figure 1—figure supplement 1D*). To assess if loss of *minar2* resulted in genetic compensation (*El-Brolosy et al., 2019*; *Ma et al., 2019*) by upregulating *kiaa1024/ubtor/minar1* gene expression, we carried out quantitative PCR and found that expression levels of *ubtora/minar1a* and *ubtorb/minar1b* were not changed in the *minar2*[fs139] nor the *minar2*[fs140] mutants (*Figure 1—figure supplement 1E*). We subsequently focused our investigation using the *minar2*[fs139] allele, and in some experiments we corroborated our findings using the *minar2*[fs140] allele.

We first examined the short-latency C-start (SLC) response evoked by auditory stimuli (*Burgess and Granato, 2007*; *Wolman et al., 2011*) to assess the hearing abilities of zebrafish larvae. We found the SLC response rates to a 200 Hz stimulus were significantly reduced in the *minar2*[fs139] mutant at 8 dpf (*Figure 1B*, median response rates: 80% in the wild type, and 60% in the mutant. $P<0.0001$, Mann-Whitney test).

To further analyze the hearing sensitivity of zebrafish larvae, we followed a procedure similar to the auditory brainstem response recording (*Higgs et al., 2003*; *Higgs et al., 2002*; *Wang et al., 2015*), and recorded auditory evoked potentials (AEP) in 7–8 dpf zebrafish larvae (*Figure 1C*). Two-way repeated-measures ANOVA revealed that the *minar2*[fs139] mutant had significantly elevated AEP thresholds compared with the wild type control (For genotype factor, $F(1, 22)=7.457$, $p=0.0122$. For genotype x frequency, $F(5, 110)=6.072$, $p<0.0001$). The AEP thresholds were significantly higher at 100–400 Hz tone bursts in the *minar2*[fs139] mutant (for 100 Hz tone, $151.0\pm1.2$ dB for *minar2*[fs139],

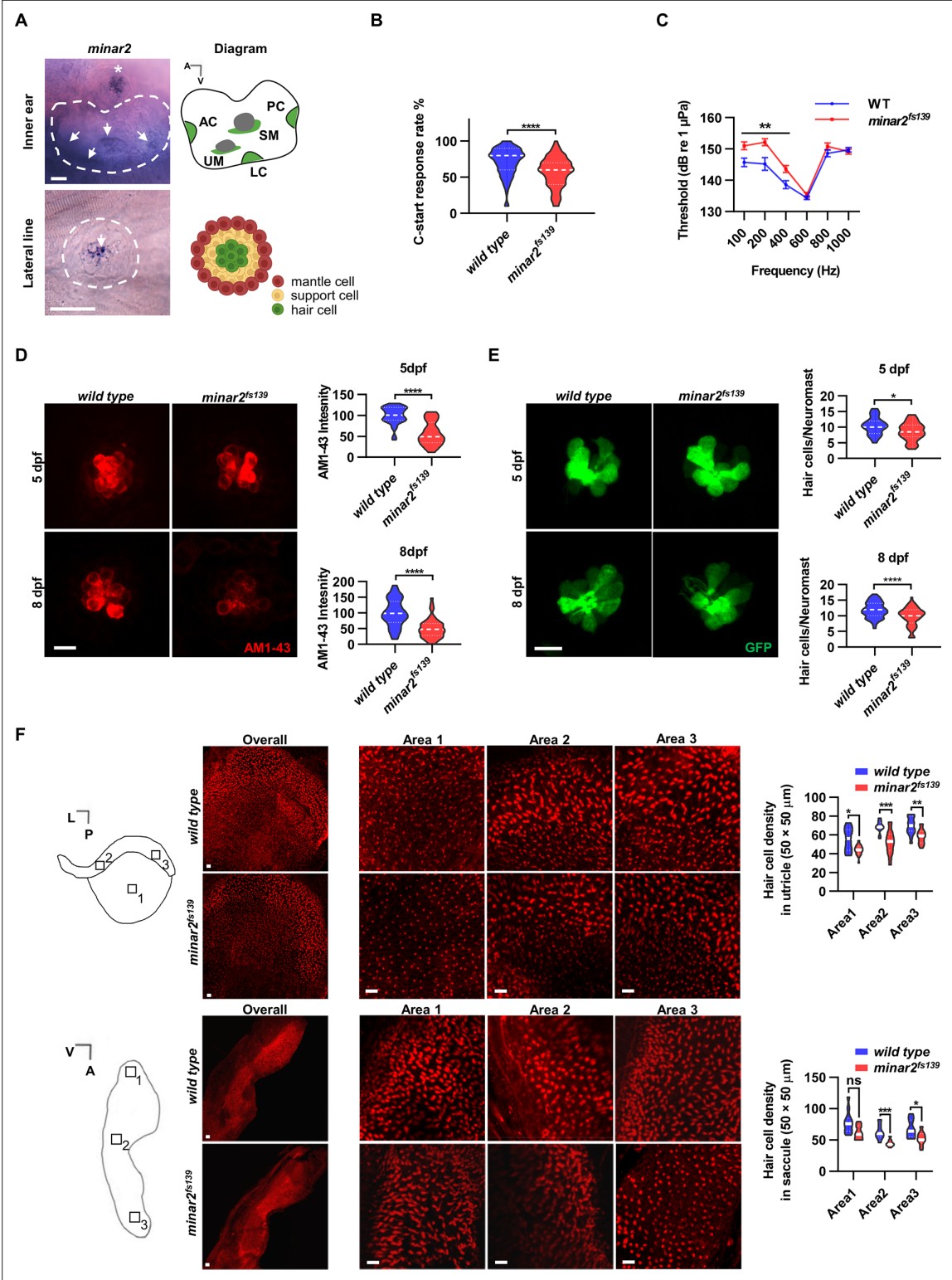

**Figure 1.** *minar2* is required for normal hearing in the zebrafish. (**A**) RNA in situ hybridization results showed that *minar2* was specifically expressed by the hair cells of the inner ears and the lateral line neuromasts (5 dpf). Arrows point to hair cells. An asterisk in the upper panel marks a head neuromast located next to the inner ear. AC: anterior crista; LC: lateral crista; PC: posterior crista; UM: utricular macula; SM: saccular macula. (**B**) C-start response rates for wild type and homozygous *minar2^{fs139}* mutants at 8 dpf (n=63 and 64, respectively. ****p<0.0001, Mann-Whitney test). (**C**) Auditory evoked potentials (AEP) thresholds in wild type and the *minar2^{fs139}* mutants (n=11 and 13, respectively. For 100-, 200-, and 400 Hz, **p<0.01). (**D**) Evaluation of mechanotransduction by AM 1–43 staining. The lateral line L3 neuromasts of 5 dpf and 8 dpf larvae were imaged and quantified (for 5 dpf, n=35 and

*Figure 1 continued on next page*

*Figure 1 continued*

36, t=7.465, df = 64.84, ****p<0.0001; for 8 dpf, n=49 and 51, t=6.444, df = 86.90, ****p<0.0001). (**E**) Quantification of hair cell numbers by counting the *myo6:Gal4FF;UAS-EGFP*-positive cells in lateral line L3 neuromast (for 5 dpf, n=30 and 32, t=2.578, df = 59.93, *p=0.0124; for 8 dpf, n=58 and 58, t=4.148, df = 114, ****p<0.0001). (**F**) Quantification of hair cell numbers in the inner ears of zebrafish adult. Hair bundles in dissected utricles (upper panels) and saccules (lower panels) were labeled with fluorescence-conjugated phalloidin. Diagrams of a utricle and saccule on the left. Numbered boxes (1-3) in the diagrams indicate the positions of imaged and counted areas (for utricles, n=15 and 19; for saccules, n=12 and 9. *p<0.05, **p<0.01, ***p<0.001). A: anterior; L: lateral; P: posterior; V: ventral. Scale bars represent 25 µm (**A**), and 10 µm (D, E, and F).

The online version of this article includes the following source data and figure supplement(s) for figure 1:

**Source data 1.** Functional requirement and expression of minar2 in hair cells *Figure 1B-F**Figure 1—figure supplement 1B, D, E*; *Figure 1—figure supplement 2A-C*.

**Figure supplement 1.** Expressions of *minar2* orthologs in hair cells and generation of *minar2* mutant alleles in the zebrafish.

**Figure supplement 2.** Numbers of inner ear hair cells in zebrafish larvae and adults.

145.7±1.4 dB for wild type control; for 200 Hz tone, 152.2±1.1 dB for *minar2^fs139^*, 145.2±2.0 dB for wild type control; for 400 Hz, 143.6±1.1 dB for *minar2^fs139^*, 138.6±1.3 dB for wild type control). These results indicate that *minar2* is required for normal hearing in the zebrafish larvae, and together with the loss of hearing phenotype in *Minar2* knockout mice (*Bowl et al., 2017*; *Ingham et al., 2019*), strengthen the conclusion that *minar2* orthologs play essential roles in the auditory functions of the vertebrates.

## AM1-43 labeling is reduced in *minar2* mutant

Because *minar2* was specifically expressed by the mechanosensitive hair cells, we examined whether there were defects in the hair cells in the *minar2* mutants. We first determined levels of mechanotransduction in zebrafish larvae using AM1-43 labeling (*Figure 1D*). AM1-43, a fixable analog of FM1-43, is a vital fluorescence dye that labels hair cells by traversing open mechanosensitive channels (*Meyers et al., 2003*), thus providing an estimate of active mechanotransduction. We found the labeling intensities of AM1-43 were markedly reduced in the hair cells of the lateral line neuromasts of the *minar2^fs139^* mutant (56.8% and 51.1% of wild-type controls at 5 dpf and 8 dpf, respectively). We next counted the number of hair cells in the lateral line neuromasts by crossing the *minar2^fs139^* mutant with a *myo6:Gal4FF;UAS-EGFP* transgenic line, which specifically labeled hair cells (*Figure 1E*). The results showed the average number of hair cells was decreased in the *minar2^fs139^* mutant (82.3% and 82.1% of wild-type controls at 5 dpf and 8 dpf, respectively). Because the decreases in hair cell numbers (~18% reduction) are smaller than those reductions of AM1-43 labeling intensities (~50% reduction), these results suggest that loss of *minar2* function mainly affects hair cell mechanotransduction in the zebrafish larvae.

We also examined the number of inner ear hair cells in the *minar2* mutant. In the zebrafish larvae, loss of *minar2* didn't alter the number of hair cells in the inner ears (for 5 dpf wild type: 23.9±0.6, *minar2^fs139^* mutant: 23.6±0.4, *minar2^fs140^* mutant: 24.9±0.6 hair cells, $F_{(2, 82)}=1.399$, p=0.253; for 8 dpf wild type: 32.5±0.7, *minar2^fs139^* mutant: 32.9±0.5, *minar2^fs140^* mutant: 33.0±0.7 hair cells, $F_{(2, 79)}=0.2378$, p=0.789, both of the lateral crista of inner ear, *Figure 1—figure supplement 2A*). The homozygous *minar2* mutants were viable, and the body length and body weight of adults were no different from the wild-type controls (*Figure 1—figure supplement 2B*). Nevertheless, hematoxylin and eosin staining of head sections suggested the inner ear regions were smaller and the numbers of hair cells were decreased in the utricle, semicircular canal crista, and saccule of the *minar2^fs139^* mutant (*Figure 1—figure supplement 2C*). To quantify the number of hair cells in adult zebrafish, we excised utricles and saccules from the inner ears and labeled the hair cells with fluorescence-conjugated phalloidin. The average numbers of hair cells were broadly decreased across different regions of utricles and saccules in the *minar2^fs139^* mutant, ranging from 71.2% to 84.3% of wild-type controls (*Figure 1F*, for the utricles, the genotype factor, $F_{(1, 32)}=37.77$, p<0.0001; for the saccules, the genotype factor, $F_{(1, 19)}=19.94$, p<0.001). We conclude that *minar2* is required for mechanotransduction in the zebrafish larvae, and loss of *minar2* reduces the number of inner ear hair cells in the zebrafish adults.

## Minar2 protein localizes to the stereocilia and the apical region of the hair cells

To study how Minar2 functioned in the hair cells, we first examined the subcellular localization of Minar2 protein in the hair cells. As the Minar2 antibodies were not available, we generated a GFP-Minar2 fusion construct driven by a hair cell-specific promoter, *myosin 6b* (*Kindt et al., 2012*; *Maeda et al., 2017*). We injected this construct into fertilized oocytes and expressed the GFP-Minar2 fusion in the hair cells. We observed that GFP-Minar2 was localized to the stereocilia and the apical region

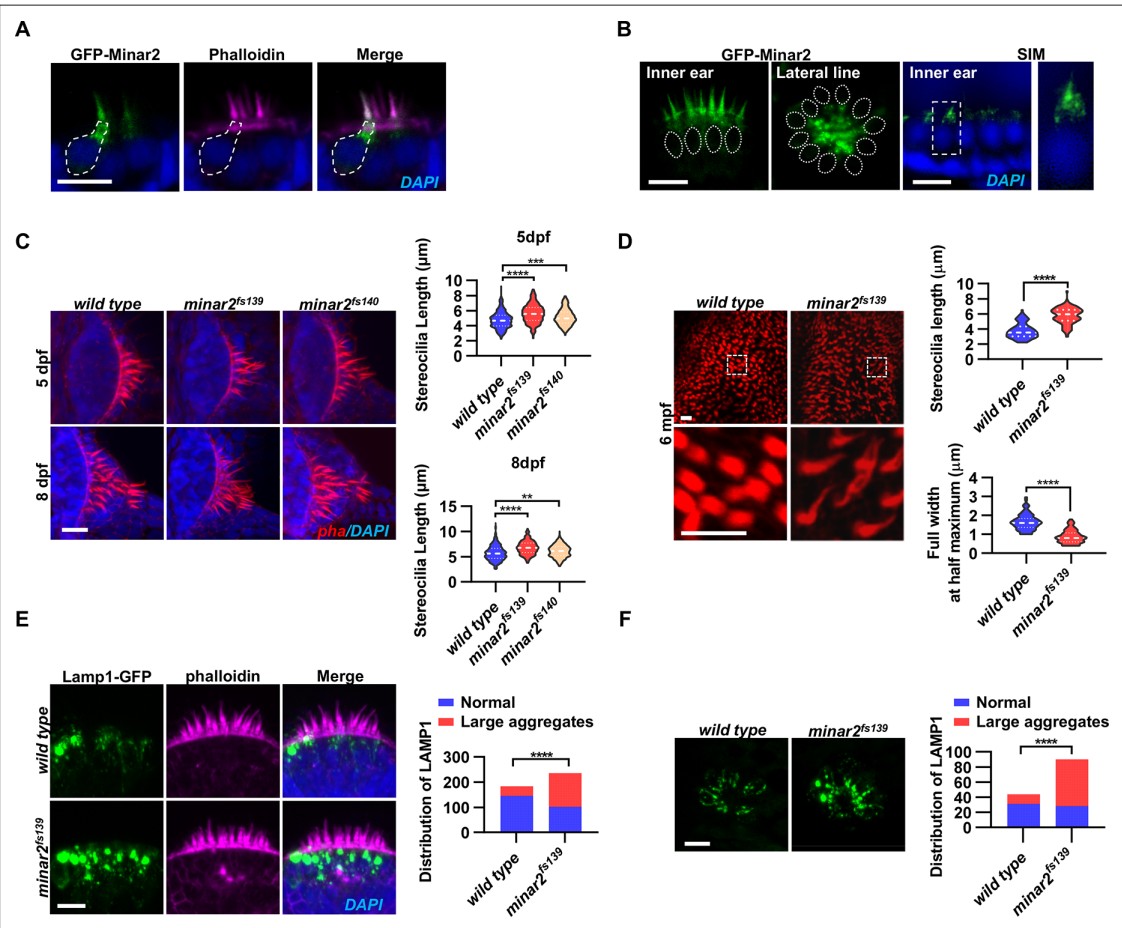

**Figure 2.** Localization and function of Minar2 in the stereocilia and the apical region of the hair cells. (**A**) Representative images of transiently expressed GFP-Minar2 in hair cells. The dashed line marks the border of a hair cell expressing GFP-Minar2. Stereocilia were labeled with phalloidin. Nuclei were counterstained by DAPI. (**B**) Distribution of GFP-Minar2 in hair cells in the stable *myo6*:GFP-Minar2 transgenic line. Representative images of hair cells of lateral crista of the inner ear (Inner ear) and lateral line neuromast (Lateral line). Dashed lines mark the nuclei of hair cells. Hair cells were also imaged with structured illumination microscopy (SIM), a super-resolution method. The right panel shows an enlarged view of the boxed area. (**C**) Quantification of hair bundle lengths of the inner ear hair cells in zebrafish larvae. Hair bundles were labeled with phalloidin and the lateral crista regions of inner ears were imaged. Hair bundle lengths were measured from 34, 29, and 22 images of wild type, *minar2^fs139^*, and *minar2^fs140^* larvae at 5 dpf, or 36, 23, and 23 images of respective larvae at 8 dpf. For 5 dpf, n=340, 290, and 220, F(2, 847)=42.58, p<0.001; For 8 dpf, n=360, 230, and 230, F(2, 817)=42.95, p<0.001. Multiple comparison significance values are indicated on the graph. (**D**) Quantification of hair bundle lengths and width of inner ear hair cells in zebrafish adult (6 mpf). The bottom panels show enlarged views of the boxed area. Hair bundles in the saccules were measured from 8 images for the wild type, and 8 images for the *minar2^fs139^* mutant. n=80 and 80. ****p<0.0001. (**E–F**) Morphology and distribution of Lamp1-labeled lysosomes in the hair cells of the inner ear (**E**) and lateral line neuromast (**F**) in zebrafish larvae (5 dpf). For the inner ear, 36 and 44 images of lateral crista regions in the wild type and *minar2^fs139^* mutant were counted, respectively (n=184 and 236, ****p<0.0001, Fisher's exact test). For the lateral line, 10 and 15 images of lateral line L3 neuromasts were counted (n=44 and 90, ****p<0.0001, Fisher's exact test). Scale bars represent 10 μm.

The online version of this article includes the following source data and figure supplement(s) for figure 2:

**Source data 1.** Localization and function of Minar2 in the apical regions of hair cells *Figure 2C-F*, *Figure 2—figure supplement 1D*.

**Figure supplement 1.** Subcellular localization of Minar2 protein.

**Figure supplement 2.** Subcellular localization of GFP-Minar2 in cultured cells.

of the hair cell, apparently around and just below the cuticular plate. The GFP-Minar2 fluorescence co-localized with the phalloidin-labeled stereocilia (*Figure 2A*). These spatial distribution patterns of GFP-Minar2 were similar regardless of apparent expression levels. To corroborate the distribution of GFP-Minar2, we generated a FLAG-tagged Minar2 construct, *myo6b*:GFP-P2A-FLAG-Minar2, which allowed labeling of hair cells by GFP and localization of Minar2 by the small FLAG tag. The results showed that FLAG-Minar2 was similarly localized to the stereocilia and the apical region of hair cells (*Figure 2—figure supplement 1A*). We subsequently generated a stable transgenic line using the *myo6b*:GFP-Minar2 construct and we found that GFP-Minar2 was also localized to the stereocilia and the apical region of the hair cells in the transgenic animals (*Figure 2B*). Prominent vesicle-like structures were seen below the cuticular plate, while a ring-like structure was often observed at the base of the hair bundle. We co-labeled the kinocilia with anti-acetylated tubulin antibodies in the *myo6b*:GFP-Minar2 transgenic zebrafish and found there was no observable GFP-Minar2 signal in the kinocilia (*Figure 2—figure supplement 1B*).

To better examine the subcellular localization, we imaged hair cells with structured illumination microscopy (SIM), a super-resolution microscopy method. The SIM results showed that GFP-Minar2 was broadly distributed in the stereocilia. Below the stereocilia and in the apical region of the hair cell, the GFP-Minar2 signal was composed of multiple small-sized vesicles of various shapes (*Figure 2B*).

## Hair bundles are longer and thinner in *minar2* mutant

Hair bundles are essential for mechanotransduction. Because the Minar2 protein localizes to the hair bundles and the apical regions where the hair bundles reside, we next examined the hair bundles in the *minar2* mutant. We labeled the hair bundles with phalloidin and observed that the hair bundles of inner ear hair cells were longer in the mutant *minar2^fs139^* and *minar2^fs140^* larvae (*Figure 2C*, for 5 dpf, wt: 4.77±0.06 µm, *minar2^fs139^*: 5.62±0.07 µm, *minar2^fs140^*: 5.15±0.07 µm, $F(2, 847)=42.58$, $p<0.001$; for 8 dpf, wt: 5.72±0.08 µm, *minar2^fs139^*: 6.80±0.09 µm, *minar2^fs140^*: 6.10±0.08 µm, $F(2, 817)=42.95$, $p<0.001$), and the hair bundles also appeared thinner. The thinning and lengthening of the mutant hair bundles were more pronounced in adult *minar2^fs139^* mutants (*Figure 2D*). In the 6-month-old adults, the average length of mutant hair bundles was more than 50% longer than that of wild type controls (wt: 3.77±0.11 µm, *minar2^fs139^*: 5.77±0.12 µm, $t=12.05$, $df = 158$, $p<0.0001$), and the width was only half that of the controls (wt: 1.66±0.05 µm, *minar2^fs139^*: 0.88±0.04 µm, $t=12.65$, $df = 158$, $p<0.0001$). We also found that the kinocilia of lateral line hair cells were disorganized in the *minar2* mutant larvae, in contrast to the normal bundled together morphology in the wild-type animals (*Figure 2—figure supplement 1D*, for 5 dpf, *minar2^fs139^*: $p<0.001$, *minar2^fs140^*: $p<0.01$; for 8 dpf, *minar2^fs139^*: $p<0.001$, *minar2^fs140^*: $p<0.01$. Fisher's exact test). The disorganized kinocilia bundle was a reminiscence of the kinocilia morphology seen in neuromasts following mechanical injury (*Holmgren et al., 2021*).

## Enlarged lysosome aggregates locate at the apical region of the hair cells in *minar2* mutant

Previous electron microscopy and immunofluorescence microscopy studies showed that the apical region of the hair cell is teeming with endocytotic vesicles, many of which are lysosomes (*Revelo et al., 2014*; *Spicer et al., 1999*; *Wiwatpanit et al., 2018*). We generated a GFP fusion construct for Lamp1, a lysosome marker, and expressed the Lamp1-GFP fusion in the hair cells using the *myo6b* promoter. Similar to the findings in the mouse model, we observed that Lamp1-GFP-positive lysosomes were abundantly distributed in the apical region of the inner ear hair cells. There were also a few lysosomes distributed along the basolateral membrane of the hair cells. In the wild-type controls, most of these Lamp1-GFP labeled lysosomes were small sphere- or rod-shaped vesicles. Occasionally, rare and large-sized Lamp1-GFP-labeled structures were observed (in 39 out of 184 hair cells) and these structures likely were abnormally enlarged lysosomes or lysosome aggregates (diameter >2 µm). In contrast, in the inner ear hair cells of the *minar2^fs139^* mutant, the large round-shaped lysosomal structures (diameter >2 µm) were frequently observed (observed in 134 out of 236 hair cells), and these abnormally enlarged lysosome aggregates were always located at the apical region of the hair cells (*Figure 2E*, $p<0.0001$, Fisher's exact test). Similar results showing abnormally enlarged lysosome aggregates were also observed in the hair cells of the lateral line neuromasts (*Figure 2F*, $p<0.0001$, Fisher's exact test). These data suggested that Minar2 may play a role in regulating the apical lysosomes in the hair cells.

Consistent with this view, GFP-Minar2 signals partially overlapped with the Lamp1-mCherry signals at the apical region of the inner ear hair cells (*Figure 2—figure supplement 1C*).

We next examined the subcellular localization of Minar2 in cultured cells in vitro. Because the *MINAR2* gene is expressed at low levels in cultured human cell lines (<1.4 nTPM in over 60 human tissue cell lines, Human Protein Atlas, proteinatlas.org), we expressed a GFP-Minar2 fusion construct in cultured HEK293 and Cos-7 cells and used KDEL, GCC1/GM130, and Lyso-Tracker to label the endoplasmic reticulum, Golgi complex, and lysosome, respectively. We found GFP-Minar2 most strongly co-localized with lysosomes (*Figure 2—figure supplement 2A–C*). Furthermore, when the morphology and distribution of the lysosomes were altered after treatment with U18666A, GFP-Minar2 signals followed the changes of lysosomes, increased in the particle sizes, and accumulated toward the nuclear region (*Figure 2—figure supplement 2D*). We further stained the lysosome lumen with filipin staining of cholesterol, which was trapped inside the lysosome after treatment with U18666A; and we observed that GFP-Minar2 signals appeared as circles that circumscribed the filipin signals, indicating GFP-Minar2 was localized on the lysosomal membranes (*Figure 2—figure supplement 2E*).

## Minar2 increases cholesterol labeling and colocalizes with cholesterol in cultured cells

Because exhaustive protein sequence homology searches failed to identify any functional domains in Minar2, we attempted sequence pattern searches (*Liu et al., 2006*) to provide clues into Minar2's function. We first extracted highly conserved sequence patterns from the multiple sequence alignment result of Minar2 protein orthologs (*Figure 1—figure supplement 1A*), then performed a protein pattern search against the Swiss-Prot database. One of the conserved sequence patterns (π-S/T-Ω-S/T-Ψ-ζ-ζ-Ω) had 125 hits in the Metazoa [taxid:33208] proteins, and 37 out of the 125 hit sequences matched to Caveolin-1 protein from various species (*Figure 3A*). A close inspection revealed that the matched sequence pattern resided in the caveolin scaffolding domain (CSD), which is known to interact with other proteins (*Murata et al., 1995*; *Razani et al., 2002*) and lipid membranes (*Schlegel et al., 1999*). The CSD of caveolin also directly binds to cholesterol in the membranes (*Ikonen et al., 2004*; *Liu et al., 2016*; *Murata et al., 1995*) and this binding may contribute to the enrichment of cholesterol in the caveolae (*Everson and Smart, 2005*; *Frank et al., 2006*).

We next investigated if Minar2 may function through cholesterol. We expressed the GFP-Minar2 fusion construct in HEK293 and Cos-7 cells, then stained these cells with filipin, a fluorescent polyene antibiotic that recognizes unesterified cholesterol in biological membranes (*Maxfield and Wüstner, 2012*; *Severs, 1997*). In the control cells expressing GFP, the filipin signals were diffusely distributed across the cells and there was no specific overlap between the GFP and the filipin signals. In contrast, in cells expressing GFP-Minar2 fusion protein, the filipin signals were accumulated in the perinuclear regions and there were strong overlaps between the Minar2 and the filipin signals in the perinuclear regions (*Figure 3B*). Quantification of the filipin signals revealed that there were significant increases of intracellular unesterified cholesterol in cells expressing GFP-Minar2 (total filipin fluorescence, for HEK293 cells, 79.0±3.0 au for GFP-only control, 106±3.8 au for GFP-Minar2 group, t=5.654, df = 104.6, p<0.0001; for Cos-7 cells, 100.5±8.2 au for GFP-only control, 140.3±12.9 au for GFP-Minar2 group, t=2.608, df = 55.73, p=0.012). The quantification results also confirmed that GFP-Minar2 recruited the filipin signals to where GFP signals were located (recruited filipin intensity, for HEK293 cells, 5.85±0.51 au for GFP-only control, 23.56±1.36 au for GFP-Minar2 group, t=12.20, df = 72.42, p<0.0001; for Cos-7 cells, 12.08±1.1 au for GFP-only control, 22.14±2.31 au for GFP-Minar2 group, t=3.903, df = 48.96, p<0.001).

Previous studies suggested that cholesterol in the biological membranes is organized in three pools, a fixed pool for membrane integrity, a sequestered pool with low chemical activity, and an active/accessible pool with high activity (*Das et al., 2014*; *Lange and Steck, 2016*; *Lange et al., 2004*). Protein probes derived from the bacterial toxin Perfringolysin O (PFO) and the minimal cholesterol-binding domain 4 (D4 or D4H) of PFO have been widely used to measure the accessible pool of cholesterol in biological membranes (*Das et al., 2014*; *Lim et al., 2019*; *Maekawa and Fairn, 2015*; *Naito et al., 2019*; *Schoop et al., 2021*). We expressed a D4H-mCherry probe to examine the effects of Minar2 on the accessible cholesterol in cultured cells. Since the D4H-mCherry protein probe was expressed in the cytoplasm, and unlike filipin the protein probe cannot cross membranes, D4H-mCherry only recognizes accessible cholesterol on the cytosolic leaflets of plasma and organelle

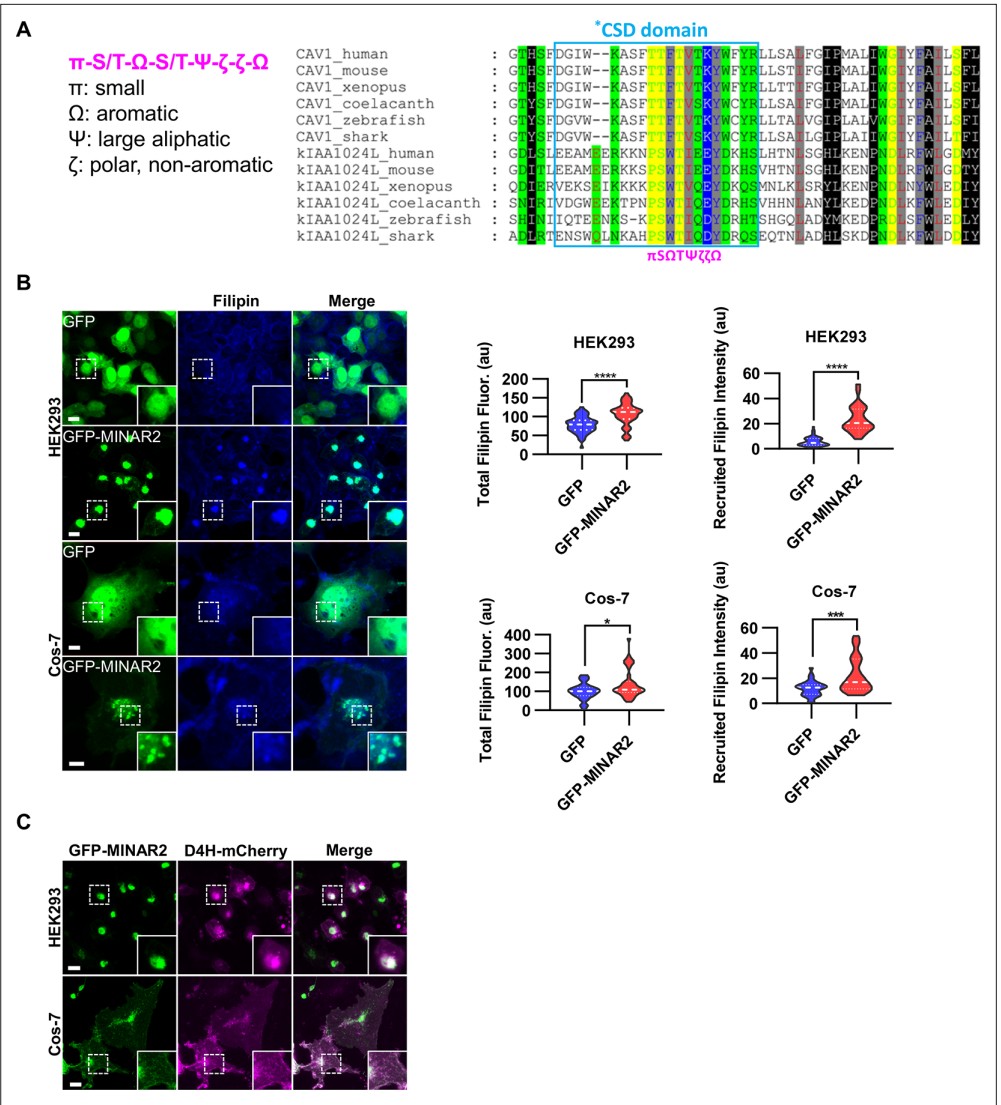

**Figure 3.** Minar2 increases cholesterol labeling and colocalizes with cholesterol in cultured cells. (**A**) Protein sequence pattern search for Minar2 identifies caveolin. The conserved Minar2 sequence pattern is written in normalized symbols (*Aasland et al., 2002*; *Livingstone and Barton, 1993*). Sequence alignment is highlighted by the physico-chemical properties of the amino acids. *CSD: caveolin scaffolding domain. (**B**) Effects of Minar2 on levels and distributions of filipin labeling in cultured cells. Total filipin fluorescence indicates the sum of all pixel values of filipin signals. Recruited filipin represents the average pixel values of filipin signals located within the GFP-positive area. For HEK293 cells, n=51 and 58; for Cos-7 cells, n=28 and 35. au: arbitrary unit. (**C**) Distribution of GFP-MINAR2 and D4H-mCherry in cultured cells. Figure inserts show enlarged views of the boxed area. Scale bars represent 10 μm.

The online version of this article includes the following source data for figure 3:

**Source data 1.** Effects of Minar2 on levels and distributions of cholesterol in cultured cells *Figure 3B*.

membranes. Similar to what was observed in cells stained by filipin, GFP-Minar2 co-localized with D4H-mCherry signals, and recruited the D4H-mCherry signals to the perinuclear regions (*Figure 3C*). We conclude that Minar2 co-localizes with cholesterol and that Minar2 may increase intracellular cholesterol levels when overexpressed in vitro.

## The hair bundle membrane is enriched for cholesterol

Previous studies showed that cholesterol is not uniformly distributed in the hair cell membranes. In isolated outer hair cells and outer hair cells in fixed cochlear tissues, cholesterol levels are higher in the

apical membranes than those in the lateral wall membranes (*Nguyen and Brownell, 1998*; *Takahashi et al., 2016*).

To examine cholesterol distribution in the hair cells of zebrafish, we first performed filipin staining on fixed zebrafish tissues. The apical regions of inner ear hair cells showed distinct filipin labeling (*Figure 4—figure supplement 1A*). The signal-to-noise ratios of filipin labeling were not high, similar to the results of filipin labeling on fixed cochlear tissues from mice (*Rajagopalan et al., 2007*). This is likely because filipin stains unesterified cholesterol in all membranous structures, and because filipin is not very stable and photobleaches rapidly, making it less suitable on fixed tissue samples.

We next investigated the intracellular distribution of cholesterol in vivo by taking advantage of the D4H-mCherry protein probe. Strikingly, when expressed in the inner ear hair cells by microinjection, D4H-mCherry specifically labeled the stereocilia regardless of its expression levels (*Figure 4A*). Low-intensity labeling of kinocilia was also observed in cells with high levels of D4H-mCherry expression, while the basolateral membranes of hair cells were not or only very faintly labeled (*Figure 4A*). To exclude possible compounding factors of expression levels of probes or special membrane structures of stereocilia, we co-expressed PM-GFP, a plasma membrane probe containing the N-terminal 10 amino acid sequence of Lyn sufficient for palmitoylation and myristoylation modification and plasma membrane targeting (*Pyenta et al., 2001*). As expected, PM-GFP uniformly labeled the basolateral membranes, together with the stereocilia and the kinocilia membranes (*Figure 4A*). We quantified and compared the distribution of PM-GFP with that of D4H-mCherry in individual inner ear hair cells. We segmented the stereocilium and basolateral cell body regions of the hair cells, and then measured the areas and the fluorescence values within the segmented regions. The results showed that although the mean area of the stereocilium region was only 16.5 ± 0.1% of that of basolateral regions, the mean PM-GFP fluorescence within the stereocilium regions was 73.7 ± 4.0% of that of the basolateral regions. The over-representation of PM-GFP signal in the stereocilium region is likely because the plasma membrane in the stereocilium region is folded multiple times to give rise to a denser plasma membrane in the stereocilium region. Strikingly, the mean D4H-mCherry fluorescence within the stereocilium regions was 567.7 ± 80.1% of that of the basolateral regions, which was 7.7 times higher than that of PM-GFP (p<0.0001, Mann-Whitney test, *Figure 4—figure supplement 1B*). Therefore, the disproportional distribution of D4H-mCherry in the stereocilium is much larger than what is expected from the distribution of the general plasma membrane probe PM-GFP, indicating that there was an enrichment of D4H-mCherry in the stereocilium membrane over the basolateral membranes. To exclude the complications that the D4H-mCherry probe may aggregate in the stereocilium region without binding to cholesterol, we constructed a non-binding D4H$^{T490G-L491G}$ variant. Previous studies have shown that the threonine–leucine pair in loop L1 of PFO domain 4 is essential for cholesterol recognition, and substitution of the T490-L491 pair with glycine residues abolishes cholesterol binding without perturbation of the overall structure of PFO (*Farrand et al., 2010*). When the D4H$^{T490G-L491G}$-mCherry variant was expressed in the hair cells, it gave rise to diffused signals in the cytoplasm only (*Figure 4B*).

In the hair cells of the lateral line neuromasts, the stereocilia were similarly strongly labeled by D4H-mCherry. There were also observable but weaker D4H-mCherry signals on the basolateral membranes of the lateral line hair cells (*Figure 4A*). These data suggested that there are high levels of cholesterol located in the stereocilia membranes, while the cholesterol levels are markedly lower in the basolateral membranes of the hair cells.

## Cholesterol labeling in the stereocilia is reduced in *kiaa1024L/minar2* mutant

Because both GFP-Minar2 and D4H-mCherry were distributed in the stereocilia (*Figure 2B* and *Figure 4A*), we asked if the two signals co-localized in the hair cells. We first generated a stable transgenic reporter line of the *myo6:D4H-mCherry* construct and crossed the D4H-mCherry report line to the stable *myo6:GFP-Minar2* line. As expected, both GFP-Minar2 and D4H-mCherry signals co-localized in the stereocilia (*Figure 4C*). Just below the stereocilia and in the apical region of hair cells, there were abundant GFP-Minar2 signals, and close inspections revealed a few D4H-mCherry-labeled structures that co-localized with GFP-Minar2 (*Figure 4C* insert). D4H-mCherry additionally labeled kinocilia, in which GFP-Minar2 signals were absent. Thus, similar to co-localization results in the cultured cells in vitro (*Figure 3C*), Minar2 co-localized with cholesterol in the stereocilia in vivo.

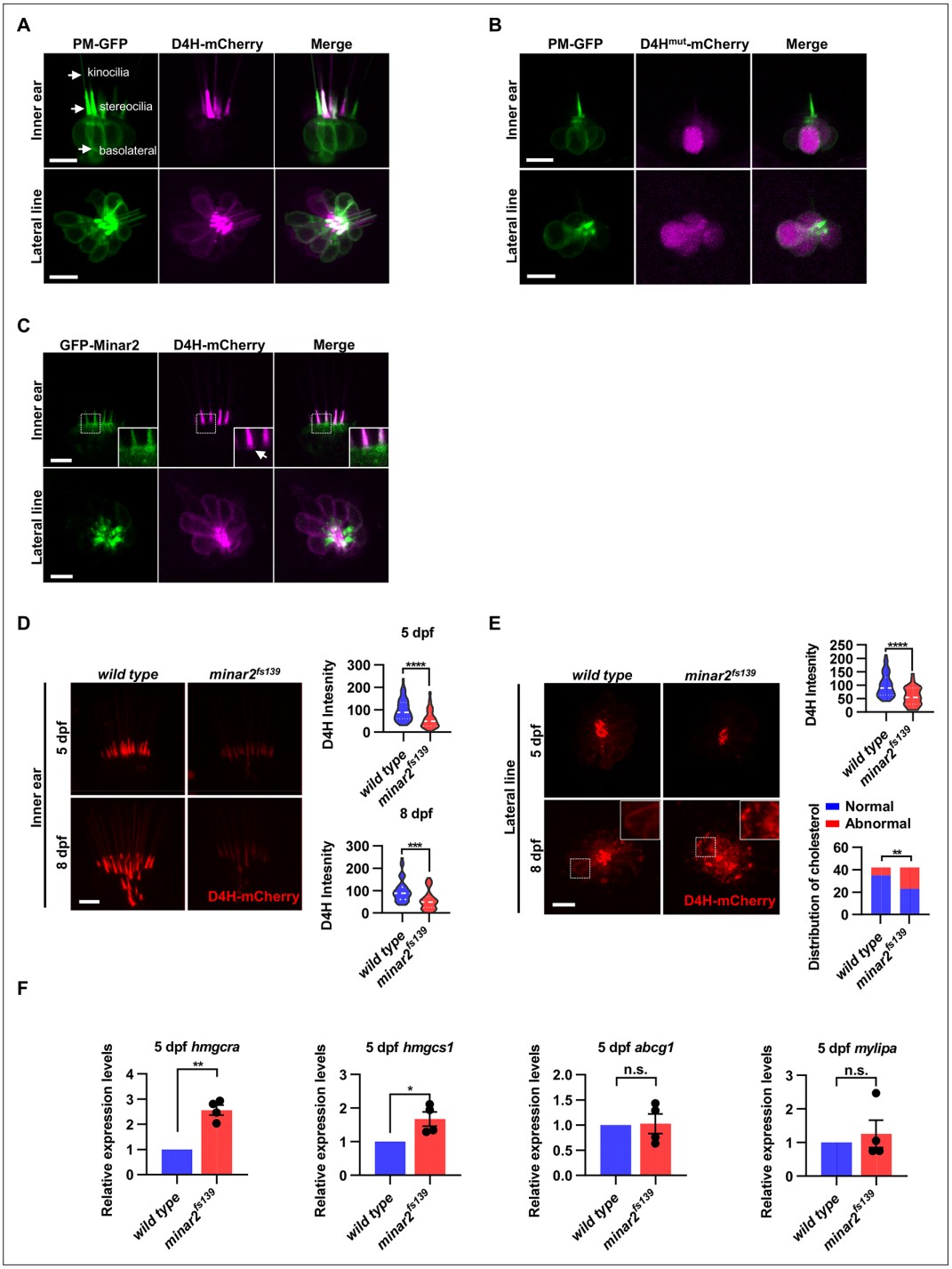

**Figure 4.** Cholesterol labeling in the stereocilia is reduced in *minar2* mutant. (**A**) Representative images of PM-GFP and D4H-mCherry expressed in hair cells. The plasma membrane probe PM-GFP labels kinocilia, stereocilia, and basolateral membranes (arrows). The cholesterol probe D4H-mCherry mostly labels the stereocilia in the inner ear. The lateral crista of the inner ear and the L3 lateral line neuromast were imaged. (**B**) Representative images of PM-GFP and non-binding D4H^mut-mCherry expressed in hair cells. The non-binding D4H^mut-mCherry carried a D4H^T490G-L491G mutation that abolishes cholesterol binding. (**C**) Distribution of GFP-Minar2 and D4H-mCherry in hair cells in stable transgenic zebrafish. GFP-Minar2 and D4H-mCherry extensively co-localize in the stereocilia and a few structures just below the stereocilia (arrow in figure insert). (**D**) Quantification of the intensity of cholesterol probe D4H-mCherry in the inner ear hair cells. The lateral crista regions of the inner ears were imaged and quantified. For the 5 dpf groups, n=49 and 49 for the wild type and the *minar2^fs139* mutant, respectively. t=4.446, df

*Figure 4 continued on next page*

*Figure 4 continued*

= 93.30, ****p<0.0001; For the 8 dpf groups, n=39 and 36, t=3.982, df = 72.30, ***p<0.001. (**E**) Quantification of the intensity and appearance of D4H-mCherry in the lateral line hair cells. The lateral line L3 neuromasts were imaged and quantified. For the 5 dpf groups, the intensity of D4H-mCherry was quantified. n=40 and 33, t=4.438, df = 70.81, ****p<0.0001. For the 8 dpf groups, the appearance of abnormally enlarged vesicles was quantified. Figure inserts show large vesicles in the basolateral regions in the *minar2ʰˢ139* mutant. n=42 and 42, **p<0.01, Fisher's exact test. (**F**) Quantification of expression levels of genes involved in cholesterol metabolism. The Srebp2 target gene (*hmgcra* and *hmgcs1)* and LXR target gene (*abcg1* and *mylipa*) were examined by qRT-PCR. Expression levels relative to GAPDH levels were normalized to the wild-type control group. For *hmgcra*, t=7.805, df = 3, **p<0.01. For *hmgcs1*, t=3.217, df = 3, *p<0.05. Scale bars represent 10 µm.

The online version of this article includes the following source data and figure supplement(s) for figure 4:

**Source data 1.** Effects of *minar2* loss-of-function on cholesterol in the apical regions of hair cells *Figure 4D–F*; *Figure 4—figure supplement 1B*.

**Figure supplement 1.** The abnormal vesicles in the *minar2* mutant were co-labeled with the lysosome.

We next examined the status of cholesterol in the *minar2ʰˢ139* mutant by crossing the D4H-mCherry report line to the mutant *minar2ʰˢ139* line. The D4H-mCherry report line had a single transgenic insertion (*Figure 4—figure supplement 1C*), and we set the crossing scheme so that the wild control and the homozygous mutant larvae both had one copy of the D4H-mCherry transgene. We then analyzed the D4H-mCherry labeling in the inner ear and neuromast whole mounts at 5 and 8 dpf. In the hair cells of the inner ears in the mutant *minar2ʰˢ139* larvae, D4H-mCherry still labeled stereocilia (*Figure 4D*), but the intensities of the signals were dramatically reduced. To quantify and compare the fluorescence signals, we used custom MATLAB scripts to count pixel intensity values using optical sections through the hair cells. The results showed the average D4H-mCherry signals of inner ear stereocilia in the *minar2ʰˢ139* larvae were about half of those in the wild controls (59.16 ± 5.92% and 56.62 ± 7.15% for 5 and 8 dpf, respectively). The decreases in D4H-mCherry signals were not due to a reduction of hair cell numbers, since there were no significant differences in the numbers of inner ear hairs at these developmental stages (*Figure 1—figure supplement 2A*). The decreases were not due to changes in transgene expression either, since the western blotting results showed that the expression levels of D4H-mCherry were similar between the wild-type controls and the mutant *minar2ʰˢ139* larvae (*Figure 4—figure supplement 1C*).

In the hair cells of lateral line neuromasts, there was a similar reduction of D4H-mCherry signal at 5 dpf (59.30 ± 5.98% of controls), while the signal intensity was no different from the wild-type controls at 8 dpf (121.39% of controls, n=41 and 44 for the wild type and the *minar2ʰˢ139* mutant, respectively. t=0.8889, df = 83, p=0.3766). The discordancy between 5 and 8 dpf was also evident in the appearances of the D4H-mCherry signals. While D4H-mCherry primarily labeled the stereocilia membranes in both wild-type controls and the *minar2ʰˢ139* mutant at 5 dpf, D4H-mCherry labeled many large, round- and rod-shaped vesicles in the cell body regions of the neuromast hair cells in the *minar2ʰˢ139* mutant at 8 dpf (*Figure 4E*, p<0.01, Fisher's exact test). These D4H-mCherry-labeled large vesicles in the hair cells of *minar2ʰˢ139* mutant likely corresponded to enlarged lysosomes, as these abnormal vesicles were co-labeled by the lysosome probe Lamp1-GFP, but not by the plasma membrane probe PM-GFP (*Figure 4—figure supplement 1D*).

Because the accessible cholesterol pool recognized by the D4H probe has high activity biochemically (*Lange and Steck, 2016*; *Lange et al., 2004*; *Lim et al., 2019*), we wondered whether the reduction of accessible cholesterol in the *minar2ʰˢ139* mutant may have consequences for the cholesterol metabolism. We examined expressions of genes downstream of Srebp2 and LXR, two master regulators of sterol synthesis and turnover (*Aqul et al., 2011*; *Liu et al., 2009*). We found that there were significant increases in relative mRNA expression for Srebp2 target genes *hmgcr* (2.6±0.2 fold) and *hmgcs1* (1.7±0.2 fold) in the *minar2ʰˢ139* mutant, while expression levels of LXR target gene *abcg1* and *mylip* were not changed (*Figure 4F*). These results suggested that the sterol synthesis pathway was activated due to a reduction of the accessible cholesterol pool in the *minar2ʰˢ139* mutant larvae. We conclude that *minar2* is required for normal cholesterol distribution and homeostasis in the hair cells.

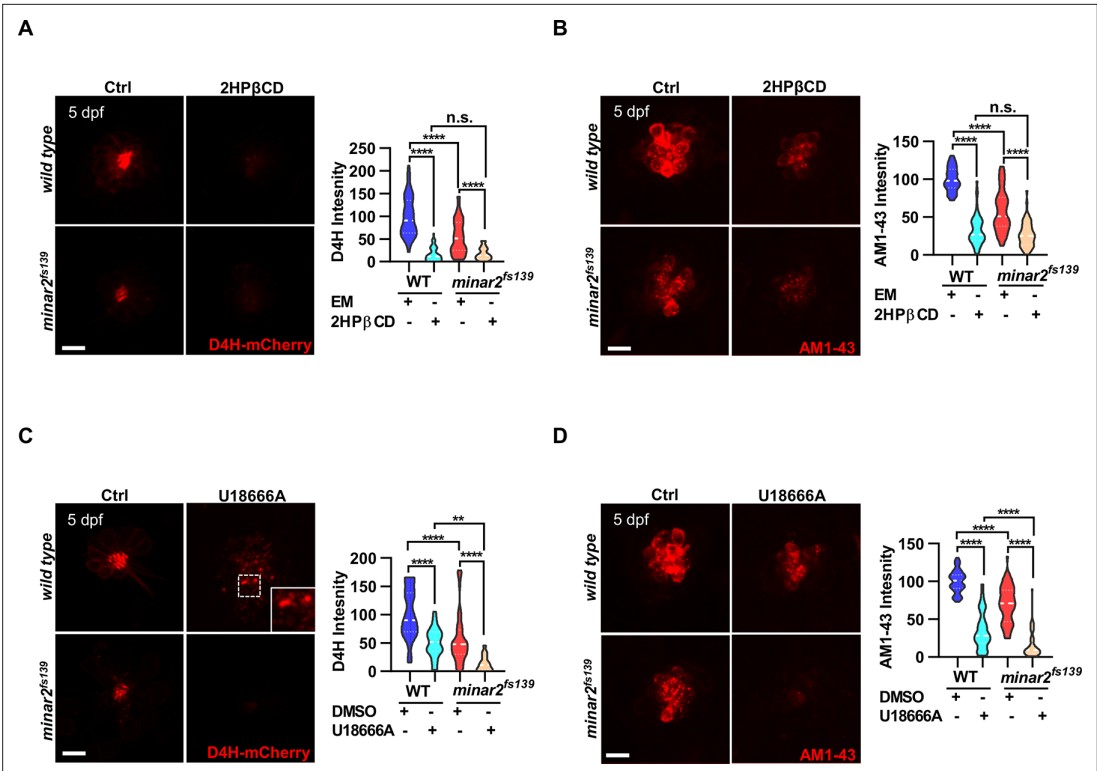

**Figure 5.** Lowering cholesterol levels aggravates hair cell defects in *minar2* mutants. (**A**) Quantification of D4H-mCherry intensity in wild type and mutant *minar2^fs139^* larvae after 2HPβCD treatment. The lateral line L3 neuromast was imaged and quantified. n=45, 41, 52, and 51. F(3, 110.2)=74.76, p<0.0001. (**B**) Quantification of AM1-43 labeling in wild type and mutant *minar2^fs139^* larvae after 2HPβCD treatment. n=49, 46, 49, and 51. F(3, 155.4)=124.4, p<0.0001. (**C**) Quantification of D4H-mCherry intensity in wild type and mutant *minar2^fs139^* larvae after U18666A treatment. n=40, 31, 42, and 33. F(3, 111.3)=39.34, p<0.0001. (**D**) Quantification of AM1-43 labeling in wild type and mutant *minar2^fs139^* larvae after U18666A treatment. n=53, 50, 52, and 46. F(3, 169.9)=158.0, p<0.0001. EM: embryonic medium, solvent control groups for 2HPβCD treatment (**A and B**); DMSO: solvent control groups for U18666A treatment (**C and D**); Multiple comparison significance values are indicated on the graph. Scale bars represent 10 µm.

The online version of this article includes the following source data and figure supplement(s) for figure 5:

**Source data 1.** Effects of decreasing cholesterol levels on hair cells in *minar2* mutant *Figure 5A–D*, *Figure 5—figure supplement 1A–B*.

**Figure supplement 1.** Effects of lowering cholesterol levels on hearing and neuromast hair cell numbers.

## Lowering cholesterol levels aggravates hair cell defects in *minar2* mutant

To further examine the involvement of cholesterol in *minar2* loss of function, we manipulated intracellular cholesterol levels using pharmacological treatments. Since the hair cells of lateral line neuromasts are superficially located, making it straightforward to apply pharmacological agents, we focused our investigation on the neuromast hair cells. We first used 2-hydroxypropyl-β cyclodextrin (2HPβCD), an oligosaccharide and a cholesterol-chelating reagent widely used to extract cholesterol from live cells (*Kilsdonk et al., 1995*; *Zidovetzki and Levitan, 2007*). As expected, treatment with 2HPβCD markedly decreased D4H-mCherry signals in the neuromast hair cells, indicating the cholesterol levels were reduced (16.5 ± 2.4% of those of control, *Figure 5A*). Moreover, the AM1-43 labeling intensities were significantly reduced after the 2HPβCD treatment (33.5% of those of controls), indicating the mechanotransduction was compromised when intracellular cholesterol was extracted (*Figure 5B*). The AM1-43 labeling intensities in the 2HPβCD treatment group were lower than those in the *minar2^fs139^* mutant (2HPβCD treatment group: 33.5 ± 3.0%, and *minar2^fs139^* mutant: 58.0 ± 3.9% of those of controls), likely because the cholesterol levels were also lower in the 2HPβCD treatment group (2HPβCD treatment group: 16.5 ± 2.4%, and *minar2^fs139^* mutant: 53.9 ± 5.2% of those of controls). When the *minar2^fs139^* mutants were treated with 2HPβCD, the cholesterol levels in the hair bundles and the AM1-43 labeling intensities were further reduced to levels in the 2HPβCD-treated wild-type controls (*Figure 5A–B*). Examination of the short-latency C-start response showed that 2HPβCD

treatment had similar effects as it did on the AM1-43 labeling. It greatly compromised hearing in the wild-type larvae, and reduced hearing further in the *minar2^fs139* mutants (**Figure 5—figure supplement 1A**). These data indicated that decreasing cholesterol levels in hair cells by 2HPβCD mimicked the defects in the *minar2^fs139* mutant. In addition, 2HPβCD treatment aggravated hair cell defects in the *minar2^fs139* mutant.

We next used U18666A, a cationic amphiphile that inhibits Niemann-Pick C1 (NPC1) protein and cholesterol egress from lysosomes (**Lu et al., 2015**). Unlike a general reduction of cholesterol levels after 2HPβCD treatment, treatment with U18666A caused a strong accumulation of cholesterol in the lysosome lumen in cultured cells (**Figure 2—figure supplement 1E**). Nevertheless, the D4H-mCherry labeling intensities were significantly decreased after the U18666A treatment (50.2 ± 4.6% of those of control, **Figure 5C**). This was due to most cholesterol molecules being trapped inside the lysosome and no longer accessible for D4H-mCherry, which was located in the cytoplasm. Treatment with U18666A also altered the distribution of the D4H-mCherry signals and gave rise to D4H-mCherry-labeled endo-membrane particles at basolateral regions in the hair cells (**Figure 5C**, figure insert). Similar to the 2HPβCD treatment group, U18666A treatment strongly decreased the intensities of AM1-43 labeling, to levels similar in the 2HPβCD treatment group (**Figure 5D**). When the *minar2^fs139* mutant was treated with U18666A, the cholesterol levels and the AM1-43 labeling intensities were further reduced to significantly lower levels than those in the U18666A-treated wild-type controls (**Figure 5C–D**). Thus, there was an additive effect between the U18666A treatment and the *minar2^fs139* mutation. We also counted the number of hair cells in the lateral line neuromasts after treatments with 2HPβCD and U18666A. The results showed both 2HPβCD and U18666A caused small reductions in hair cell numbers in wild-type controls, but only U18666A further reduced the numbers of hair cells in the neuromasts of mutant *minar2^fs139* larvae (**Figure 5—figure supplement 1B**).

## Increasing cholesterol levels rescue hair cell defects and hearing in *minar2* mutant

We then tested whether increasing cholesterol levels had any effect on the hair cells. We first attempted treating wild-type larvae with a 2HPβCD/cholesterol complex and found the 2HPβCD/cholesterol complex precipitated out in the medium, likely due to the incubation temperature of 28.5 °C for proper zebrafish development (data not shown). We next used efavirenz, a cholesterol 24S-hydroxylase (CYP46A1) inhibitor, which can increase intracellular cholesterol levels by inhibiting hydroxylation of cholesterol and subsequent removal of cholesterol from the cell (**Petrov and Pikuleva, 2019**). We found that efavirenz consistently increased cholesterol levels in the neuromast hair cells. We observed that treating the wild-type larvae with efavirenz caused small increases in D4H-mCherry signals (111.2 ± 11.2% of controls, p=0.807) and AM1-43 labeling (107.4 ± 3.6% of controls, p=0.400) in neuromast hair cells, likely because the cholesterol homeostasis mechanism was intact in the wild type. In contrast, after treatment with efavirenz, there were significant increases of D4H-mCherry signals in the hair bundles (**Figure 6A**, from 55.8±7.0%–88.7 ± 8.4% of those of controls, p=0.021), increases of AM1-43 labeling in hair cells in the 5 dpf *minar2^fs139* mutant (**Figure 6B**, from 61.8±2.9%–96.1 ± 3.1% of those of controls, p<0.0001), and both were restored to the levels observed in the wild type controls (p=0.269 and p=0.115). We also found the treatment with efavirenz rescued the abnormally enlarged vesicles seen in the 8 dpf *minar2^fs139* mutant (**Figure 6C**). We further tested voriconazole, another CYP46A1 inhibitor, and observed that the decreased AM1-43 labeling (**Figure 6—figure supplement 1A**) and the abnormal distribution of D4H-mCherry (**Figure 6—figure supplement 1B**) were similarly reversed after the voriconazole treatments.

We then examined whether efavirenz treatment rescued hearing defects in the *minar2^fs139* mutant. We found that the compromised short-latency C-start (SLC) response rates in the *minar2^fs139* mutant were rescued to levels in the wild-type controls with the efavirenz treatment (**Figure 6D**). Thus, both the mechanotransduction (AM1-43 staining) and the hearing were restored after the decreased cholesterol levels were restored by the treatment with efavirenz.

## Minar2 interacts with cholesterol in vitro

The pharmacological treatment studies above strongly suggested that Minar2 may function through the regulation of cholesterol. We next examined whether Minar2 interacted with cholesterol directly. The CSD of caveolin binds cholesterol through a so-called CRAC motif (**Epand et al., 2005**; **Le Lan**

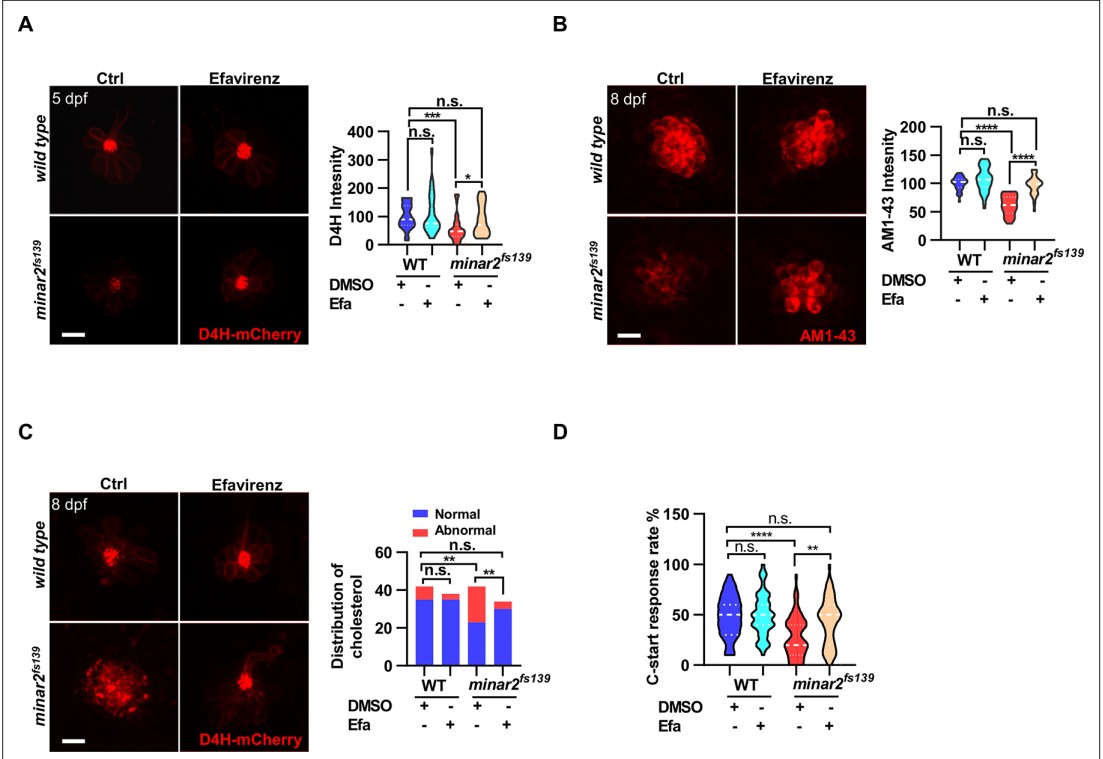

**Figure 6.** Increasing cholesterol levels rescue hair cell defects and hearing in *minar2* mutants. (**A**) Quantification of D4H-mCherry in hair cells of wild type and *minar2*$^{fs139}$ zebrafish after efavirenz treatment. The lateral line L3 neuromast was imaged and quantified. n=40, 41, 42, and 41. F(3, 127.8)=7.557, p<0.001. (**B**) Quantification of AM1-43 labeling in wild type and mutant *minar2*$^{fs139}$ larvae after efavirenz treatment. n=31, 33, 33, and 30. F(3, 115.7)=46.65, p<0.0001. (**C**) Effects of efavirenz treatment on the appearance of abnormally enlarged vesicles in hair cells of wild type and mutant *minar2*$^{fs139}$ zebrafish. n=42, 38, 42, and 34. $\chi^2$=20.92, df = 3, p<0.001. (**D**) Effects of efavirenz treatment on C-start response rates for wild type and *minar2*$^{fs139}$ mutants (n=48 for all 4 groups. p<0.0001, Kruskal-Wallis test). DMSO: solvent control groups; Efa: efavirenz treatment groups. Multiple comparison significance values are indicated on the graph. Scale bars represent 10 μm.

The online version of this article includes the following source data and figure supplement(s) for figure 6:

**Source data 1.** Effects of increasing cholesterol levels on hair cells in *minar2* mutant **Figure 6A–D**; **Figure 6—figure supplement 1A–B**.

**Figure supplement 1.** Increasing cholesterol levels by voriconazole rescue hair cell defects in *minar2* mutant.

*et al., 2010*; *Liu et al., 2016*). We identified three CRAC (and its analog CARC) motifs in the primary sequences of Minar2 orthologs (**Figure 7A** and **Figure 7—figure supplement 1A**). We reasoned that if Minar2 interacted with cholesterol directly with the CRAC/CARC motif, point mutations of the critical aromatic residues (Y/F/W) within these motifs to alanine residues (*Fantini et al., 2016*; *Li and Papadopoulos, 1998*; *Paschkowsky et al., 2018*) should compromise the association between Minar2 and cholesterol. We used the HEK293 cell to test this hypothesis in vitro, as GFP-Minar2 expressed in the cell strongly recruited cholesterol to the perinuclear region and also increased intracellular cholesterol labeling (**Figure 3B**). We transfected HEK293 cells with wild type (MINAR2$^{1-190}$) and point mutation construct (MINAR2$^{YW-A}$), then stained the cholesterol with filipin (**Figure 7B**). We observed that the point mutation of MINAR2$^{YW-A}$ compromised the recruitment of cholesterol to the perinuclear region (decreased from 24.58±1.30 au in the MINAR2$^{1-190}$ group to 14.85±0.90 au in the MINAR2$^{YW-A}$ group, p<0.0001) and the intensity of intracellular cholesterol labeling (from 103.9±4.86 au in the MINAR2$^{1-190}$ group to 86.43±3.21 au in the MINAR2$^{YW-A}$ group, p<0.01). These effects were not due to changes in expression levels of the point mutation construct, since it was expressed at a similar level as the wild-type construct (**Figure 7C**).

In addition, the recently released MINAR2 structure model from AlphaFold2 (*Tunyasuvunakool et al., 2021*) allowed us to carry out computational docking to study probable Minar2-cholesterol interactions. We found the docking site with the best binding free energy (–8.1 kJ/mol) for cholesterol was indeed located in the conserved sequence pattern that matched the CSD of caveolin, and

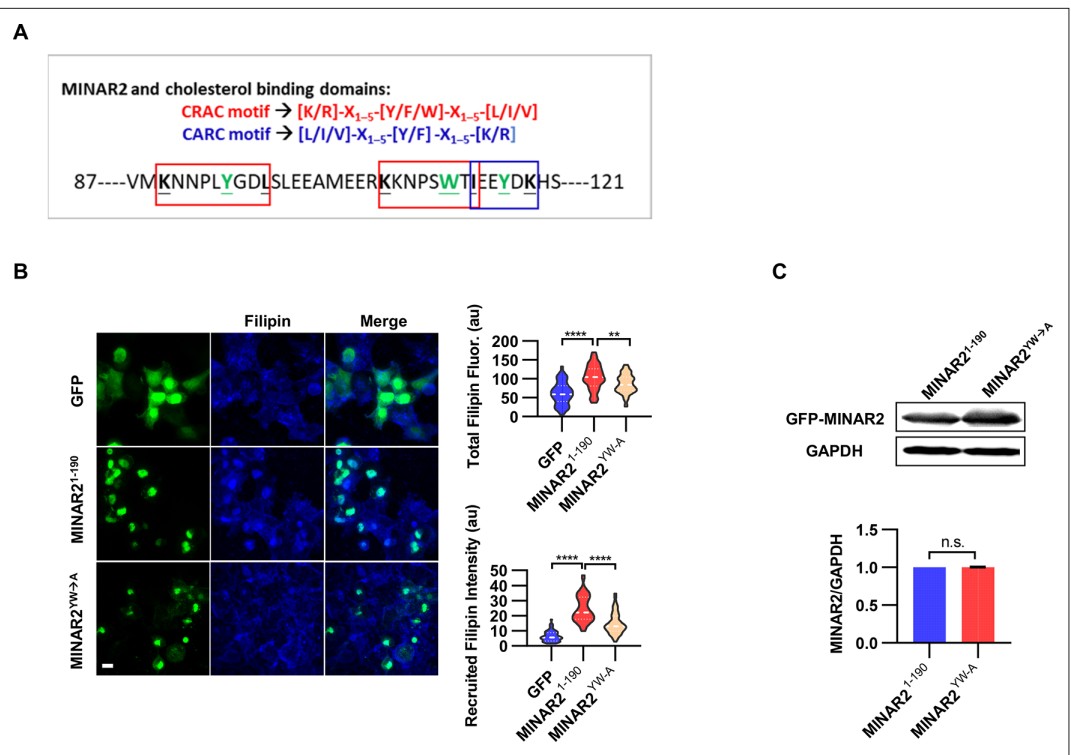

**Figure 7.** Minar2 interacts with cholesterol in vitro. (**A**) Cholesterol recognition motifs in Minar2 primary sequence. The sequences for cholesterol-recognizing amino-acid consensus (CRAC, red letter and box) and its analog CARC (blue letter and box) are indicated. The critical aromatic resides (Y/W, in green letters) were mutated to alanine in the point mutation MINAR2$^{YW-A}$ construct. (**B**) Effects of critical aromatic reside mutation on the levels and distributions of filipin labeling in cultured cells. HEK293 cells were transfected with GFP alone (GFP), full length (GFP-MINAR2$^{1-190}$), or the point mutation construct (GFP-MINAR2$^{YW-A}$). n=44, 46, and 62, for total filipin fluorescence, $F_{(2, 149)}=23.80$, p<0.0001; for recruited filipin intensity, $F_{(2, 105.5)}=79.28$, p<0.0001. Multiple comparison significance values are indicated on the graph. Scale bars represent 10 μm. (**C**) Immunoblot analysis of the expression levels of the full-length construct (GFP-MINAR2$^{1-190}$) and the point mutation construct (GFP-MINAR2$^{YW-A}$) in HEK293 cells. Expression levels relative to GAPDH were quantified. t=0.7498, df = 2, p=0.532.

The online version of this article includes the following source data and figure supplement(s) for figure 7:

**Source data 1.** Interaction between Minar2 and cholesterol *Figure 7B–C*.

**Figure supplement 1.** Computational docking between MINAR2 structure model and cholesterol.

the mutated aromatic residues (W112 and Y117) above were involved in the interaction between Minar2 and cholesterol (*Figure 7—figure supplement 1*). We next carried out docking studies with cholesterol analogs including desmosterol (MHQ, a precursor of cholesterol), 25-hydroxy cholesterol (HC3, an oxysterol), cholesterol-3-sulfate (C3S, a head modification of cholesterol), and estradiol (EST, a steroid). Except for cholesterol-3-sulfate, these cholesterol analogs interacted with MINAR2 in the same way that cholesterol did. The binding free energy was comparable for 25-hydroxy cholesterol (–8.0 kJ/mol, versus –8.1 KJ/mol for cholesterol), but was reduced for desmosterol (–6.5 kJ/mol) and estradiol (–7.4 kJ/mol). When compared to other cholesterol analogs, the docked orientation of cholesterol-3-sulfate was flipped upside down, and the binding free energy was mostly reduced (–6.1 kJ/mol).

## Discussion

We provide evidence that Kiaa1024L/Minar2 is required for normal hearing in the zebrafish. Our results suggest that the Kiaa1024L/Minar2 protein, located on the apical endo-membranes and stereocilia membranes, is responsible for the apical distribution of cholesterol to the hair bundle, and this apical distribution of cholesterol is essential for the normal morphogenesis and function of the

hair bundle. Consistent with this model, loss of *minar2* markedly reduces cholesterol distribution in the hair bundles, causes longer and thinner hair bundles, and gives rise to enlarged and aggregated apical lysosomes. These defects in *minar2* mutant likely compromise mechanotransduction, reduce the number of hair cells, and result in hearing loss. Our model is further supported by results from pharmacological interventions. The effects of *minar2* loss-of-function are mimicked by drug treatment that reduces the cholesterol levels in the hair bundles, while these effects are reversed by treatment that increases the cholesterol levels.

*Kiaa1024L/Minar2* is a previously understudied gene and is present in vertebrate species only. Previous mouse genetic screening for hearing loss genes showed that *Kiaa1024L/Minar2* knockout causes profound deafness (*Bowl et al., 2017*; *Ingham et al., 2019*). Our study presented here, together with the results in the mouse model, highlights that *Kiaa1024L/Minar2* is essential for normal hearing in vertebrates. Our results with mutant zebrafish reveal that hearing defects exist at the level of hair cells. This conclusion is consistent with the highly enriched expression of *Kiaa1024L/Minar2* in the hair cells in the vertebrate inner ears, as shown by our in situ hybridization results and our surveys of FACS and single-cell RNA sequencing data (*Barta et al., 2018*; *Elkon et al., 2015*; *Erickson and Nicolson, 2015*; *Liu et al., 2018*; *Steinhart et al., 2022*). Outside the inner ear, *Kiaa1024L/Minar2* is also expressed in various regions in the brain tissue, and a previous study showed that loss of *Minar2* in mice impairs motor function and reduces the number of tyrosine hydroxylase-positive neurons (*Ho et al., 2020*).

The degree of hearing loss in mutant *minar2$^{fs139}$* zebrafish larvae is not as severe as it was in the adult knockout mouse. A recently published study showed that *Minar2* knockout mice had degeneration of hair cells and progressive sensorineural hearing loss (*Bademci et al., 2022*). In mutant *minar2$^{fs139}$* zebrafish, the reduction of inner ear hair cell numbers is progressive, worsening from no changes in the larval stage to a 30% reduction in the adult stage. We note that unlike the mouse model, in which the hair cells are generated in early development and cannot regenerate in the adult stage (*Atkinson et al., 2015*; *Basch et al., 2016*; *Driver and Kelley, 2020*), in zebrafish the hair cells are continuously generated until late in adulthood (*Wang et al., 2015*). Additional alternatives that may account for the less severe phenotype in the zebrafish include genetic compensation (*El-Brolosy et al., 2019*; *Ma et al., 2019*). Although the expression levels of *minar1a* and *minar1b* were not changed in mutant *minar2$^{fs139}$* larvae, changes in other genes outside the UPF0258 gene family could partly remediate the loss of *minar2* in the zebrafish.

Defects in mechanotransduction, longer and thinner hair bundles, and enlarged apical lysosomes observed in *kiaa1024L/minar2* mutant all indicate that *Kiaa1024L/Minar2* acts primarily in the apical region and the hair bundle in hair cells. GFP or FLAG-tagged Minar2 protein is primarily localized to the apical endo-membranes and stereocilia membranes, which are the appropriate locations to exert the above functions. Previous studies showed that the apical region of the hair cell is crowded with endocytotic vesicles, including abundant lysosomes (*Krey et al., 2018*; *Revelo et al., 2014*; *Spicer et al., 1999*; *Wiwatpanit et al., 2018*) These endocytotic membrane-bound structures are thought to transport ions and other materials to and fro towards the apical entry of the hair cells (*Spicer et al., 1999*), but the functions of these structures are not well characterized. One recent study showed that loss of lysosomal mucolipins in cochlear hair cells gives rise to abnormally enlarged lysosomes at the apical region. These aberrant lysosomes are toxic to cells and cause hair cell loss (*Wiwatpanit et al., 2018*).

At the apical end of the inner ear hair cell, the distribution pattern of the cholesterol probe D4H-mCherry in the hair bundle is striking. The D4H-mCherry probe is almost exclusively localized in the hair bundle; while the plasma membrane probe PM-GFP is localized both to basolateral and apical membranes. Previous studies showed that the stereocilia membrane is densely covered with filipin-induced deformation in the freeze-fracture preparation of hair cells, indicating the presence of abundant cholesterol in the stereocilia membrane (*Forge et al., 1988*; *Forge and Richardson, 1993*). The exact function of this striking distribution pattern of cholesterol requires further investigation. Cholesterol is a major component of biological membranes, and it plays both structural and functional roles in the bio-membranes (*Maxfield and van Meer, 2010*; *Subczynski et al., 2017*). Previous studies showed that cholesterol may stiffen biological membranes (*Chakraborty et al., 2020*; *Dimova, 2014*). When cholesterol was depleted from hair cell membranes following methyl β cyclodextrin treatment, an apparent loss of structural integrity and 'floppy' hair bundles were

observed (*Purcell et al., 2011*). The biophysical properties of membranes may be more directly involved in mechanotransduction. It has been suggested that the stereocilia tip membranes can serve as a component of the gating spring of the MET channel (*Powers et al., 2014*; *Powers et al., 2012*), although this has currently been debated (*Bartsch et al., 2019*). On the other hand, cholesterol has direct or indirect effects on membrane proteins such as ion channels and G-protein-coupled receptors (*Harris, 2010*). Previous studies showed that the mechanosensitive ion channel Piezo1 interacts with cholesterol and this interaction is essential for the spatiotemporal activity of Piezo1 (*Buyan et al., 2020*; *Ridone et al., 2020*).

Our model suggests that Kiaa1024L/Minar2 regulates the distribution and homeostasis of cholesterol to ensure normal hearing. Several lines of evidence indicate Kiaa1024L/Minar2 interacts with cholesterol: (1) the highly conserved sequence pattern found in Kiaa1024L/Minar2 orthologs matches the cholesterol binding CSD domain of caveolins; (2) Kiaa1024L/Minar2 recruits cholesterol in vitro, and point mutations in the predicted cholesterol binding sites abolish the recruitment of cholesterol; (3) computational docking study with the atomic model of Kiaa1024L/Minar2 from AlphaFold2 shows reasonable binding free energy between cholesterol and the sequence pattern we identified. Nevertheless, further biochemical studies are required to demonstrate direct binding between the Kiaa1024L/Minar2 protein and cholesterol.

It is currently unclear how Kiaa1024L/Minar2 regulates the distribution and homeostasis of cholesterol in hair cells. Kiaa1024L/Minar2 may be responsible for or facilitate the apical transport of cholesterol to the hair bundles. The subcellular localization of Kiaa1024L/Minar2 in both the apical endo-membranes and the hair bundle membranes is consistent with a transporting role for Kiaa1024L/Minar2. The apical transport defects caused by the loss of Kiaa1024L/Minar2 can explain the marked decrease of accessible cholesterol levels in the hair bundles. Because the accessible cholesterol is the critical determinant for cholesterol homeostasis regulation (*Lange and Steck, 2016*; *Lange et al., 2004*), the decrease of accessible cholesterol is expected to bring about the up-regulation of genes downstream of Srebp2, which is indeed observed in the *minar2* mutant. According to this model, the observed enlarged aggregated lysosome and decreased number of hair cells are likely a result of the transportation defects. It will be important to examine the cholesterol dynamics in vivo and to define the interacting proteins of Kiaa1024L/Minar2 to illustrate the detailed mechanism for the function of Kiaa1024L/Minar2.

Kiaa1024L/Minar2 protein belongs to a protein family that also includes the Kiaa1024/Minar1/Ubtor protein. This protein family is defined by a conserved but *uncharacterized* protein family domain (UPF0258) located at the carboxy ends of the proteins. Our previous study showed that Kiaa1024/Minar1/Ubtor acts as a negative regulator of mTOR signaling through its interaction with the Deptor protein, an integral component of the mTORC complex (*Zhang et al., 2018*). Loss of the *ubtor* gene in zebrafish causes motor hyperactivity and epilepsy-like behaviors by elevating neuronal activity and activating mTOR signaling (*Wang et al., 2021*). It has also been shown that *Ubtor/MINAR1* plays a role in angiogenesis and breast cancer (*Ho et al., 2018*). Since the region in Kiaa1024/Minar1/Ubtor that interacts with the Deptor protein is missing in the Kiaa1024L/Minar2 protein, the regulation of mTOR signaling is likely a function specific to the Kiaa1024/Minar1/Ubtor protein. Because the cholesterol-binding region that we identify in this study is conserved in the Kiaa1024/Minar1/Ubtor protein, it should be interesting to investigate whether Kiaa1024/Minar1/Ubtor also plays a role in the regulation of cholesterol.

In conclusion, our data provide evidence that cholesterol plays an essential role in the hair bundle, and that Kiaa1024L/Minar2 regulates cholesterol distribution and homeostasis to ensure normal hearing. Because of the conserved requirement of *Kiaa1024L/Minar2* gene orthologs for hearing in both mice and zebrafish, and that the human ortholog *MINAR2* is also specifically expressed by the auditory hair cells (*Steinhart et al., 2022*), it will be important to examine whether *KIAA1024L/MINAR2* gene mutations may underline hearing abnormalities in humans. Indeed, while this study was under review, it was reported that mutations in *MINAR2* cause deafness in human patients (*Bademci et al., 2022*). Further studies of Kiaa1024L/Minar2's regulation of cholesterol distribution and homeostasis in the zebrafish and mammalian models will provide new insights into the biological processes underlying the morphogenesis and physiology of hair bundles and hearing loss.

## Materials and methods

### Resources and materials availability

All constructs, transgenes, codes, and reagents generated in this study are available from the Lead Contact without restriction. Further information and requests for resources and materials should be directed to and will be fulfilled by the Lead Contact, Gang Peng (gangpeng@fudan.edu.cn).

### Zebrafish strains

The animal use protocols were approved by the Fudan University Shanghai Medical College Institution Animal Care and Use Committee (130227–092, 150119–088 and 190221–147). All animals were handled in accordance with the Fudan University Regulations on Animal Experiments.

Zebrafish were maintained according to standard protocols (*Westerfield, 2007*). Lines used in this study included the AB, Tg(UAS:EGFP), Tg(*myo6b*:GAL4FF), Tg(*myo6b*:GFP-Minar2), Tg(*myo6b*:D4H-mCherry), and *minar2* mutant (*minar2^fs139^* and *minar2^fs140^*). The Tg(UAS:EGFP) line was a gift from Dr. Kawakami (*Asakawa et al., 2008*). Zebrafish larvae were obtained from natural spawning and fed paramecia beginning on 5 dpf. Larvae at 5–8 days post fertilization (dpf) of undifferentiated sex were examined.

### Cell lines

The HEK293, HeLa, and COS-7 cell lines were obtained from the Cell Bank of the Chinese Academy of Sciences (Shanghai). Cultured cells were regularly checked for the absence of mycoplasma contamination (Yeasen). HeLa cells were grown in MEM/EBSS (HyClone) medium. HEK293T and COS-7 cells were grown in DMEM/High glucose (HyClone) medium. The growth media were supplemented with 10% fetal bovine serum (FBS) (Biological Industries). For the transfection of cells, Lipofectamine 3000 (Thermo Fisher Scientific) was used.

### Generation of *minar2* mutant

To disrupt the *minar2* gene in the zebrafish, targeted lesions were introduced into the zebrafish genome through CRISPR/Cas9mediated gene modification. The gRNA targeted sequence (5'- GGAA TGTTGCCGGCTACACATGG-3') was located in the first exon of the *minar2* gene. Two *minar2* mutant lines (*minar2^fs139^* and *minar2^fs140^*) were each outcrossed with the AB line for 6 generations and then used in this study. Allele-specific primers (Key Resources Table Appendix) were used to genotype wild-type and mutant siblings.

### Constructs and transgenic lines

Plasmid constructs were generated by standard molecular cloning with primers and oligonucleotides listed in the Key Resources Table Appendix. Transgenic lines were generated using the Tol2 transposase-mediated method (*Kawakami et al., 2000*; *Urasaki et al., 2006*). Transgenic lines were outcrossed with the AB line for three generations before being used in this study. For the Tg(*myo6b*:D4H-mCherry) line, the genomic insertion site of the transgene was mapped by high-efficiency thermal asymmetric interlaced PCR (*Liu and Chen, 2007*; *Liu and Whittier, 1995*) as previously described (*Zhang et al., 2018*).

### C-start response

The C-start response was carried out as previously described (*Zhang et al., 2018*). Briefly, each well of a 4 by 4 test grid was filled with an individual larva of 5 or 8 dpf and 150 µL embryo medium. The larvae were allowed to adapt to the wells for 5 min before the start of the test. The microcontroller delivered 10 acoustic stimuli (200 Hz, 50 ms), with randomized inter-stimulus intervals (ranging from 40 to 120 seconds) to minimize the adaptation of the zebrafish. The video sequence was inspected by a person blind to the genotype condition, and C-start responses were recorded if the zebrafish performed the C-bend within 25 ms after the start of the stimulus. The C-start response rates were calculated by the number of C-starts performed out of the 10 stimuli delivered.

### Auditory evoked potential (AEP) recording

AEP recordings were carried out following procedures from published studies with modifications (*Higgs et al., 2002*; *Wang et al., 2015*). Each zebrafish larva (7–8 dpf) was mounted dorsal-side up in

1% low melt point agarose at the center of a 35 mm glass-bottom petri dish, then covered with 1 ml of distilled water. A recording electrode was inserted into the dorsal surface of the brainstem, between the two otic vesicles. A reference electrode was placed near the surface of the forebrain. Electrodes were made of insulated stainless steel (160 μm in diameter), and the dimensions of the exposed tip were approximately 80 μm in height, and 50 μm (base) to 5 μm (end) in diameter (Zhongyan, Beijing). AEP stimuli generation and signal acquisition were controlled by custom MATLAB scripts and a custom C++program. AEP stimuli consisted of Blackman-windowed tone pips (10ms in duration) of varying frequencies (100, 200, 400, 600, 800, and 1000 Hz) and were presented in descending sound levels in 2–5 dB decrements. AEP potentials were amplified by 80,000 x (TengJun Amp, Xi'an), digitized (Art Technology, Xi'an), then processed and averaged by custom MATLAB scripts.

### Mechanotransduction assay with AM1-43 labeling

Larvae were incubated in 0.2 μM AM1-43 (Biotium, California) in low calcium Ringer's solution (LCR) [140 NaCl, 2 KCl, 0.1 $CaCl_2$, 5 D-glucose, and 5 HEPES (mM)] at room temperature for 3 min and then washed in LCR for 2x5 min. The samples were then fixed in 4% paraformaldehyde for 1 hr at room temperature before storage and imaging. The lateral line neuromast L3 was imaged and the voxels within a bounding box (30x30x15 μm in 10 optical sections) containing the hair cells were processed for quantification using custom MATLAB scripts.

### Phalloidin staining

Larvae were fixed in 4% paraformaldehyde/1 x PBS for 1 hr at room temperature and then overnight at 4 °C. Fixed specimens were permeabilized in 1% Triton-X100/1 x PBS for 1 hr at room temperature, and then incubated with fluorescence-conjugated phalloidin (Beyotime, Shanghai, 1:200 in 1% Triton-X100/1 x PBS) for 2 hr at room temperature. DAPI was used to counter-stain the nuclei.

### Lyso-Tracker Red staining

To stain cultured cells, cells were grown in glass-bottom Petri dishes. Lyso-Track Red (Beyotime, Shanghai) was diluted (1:2000) in cell culture medium and preheated to 37°C before replacing the medium in the glass-bottom Petri dishes. Cells were incubated in medium containing Lyso-Track Red for 10 min at 37°C and then washed with culture medium before live imaging. To stain larvae, Lyso-Tracker Red was diluted (1:2000) in the embryonic medium (EM). Larvae were incubated in EM containing Lyso-Tracker Red for 1.5 hr at 28 °C. After the staining, larvae were rinsed three times with EM before live imaging. All staining steps were performed in the dark.

### Filipin staining

Filipin (MCE, Shanghai) stock was prepared in DMSO (50 mg/ml). To stain cultured cells, cells grown on coverslips were fixed with 4% formaldehyde/1 x PBS for 10 min at room temperature. Fixed cells were stained with freshly prepared filipin working solution (0.25 mg/ml in 1xPBS) for 15 min at room temperature. To stain larvae, larvae were fixed in 4% paraformaldehyde/1 x PBS for 2 hr at room temperature, then overnight at 4°C. Fixed specimens were incubated with filipin working solution for 2 hr at room temperature. Samples stained with filipin were imaged with a 405 nm laser. Quantification of filipin staining was carried out with custom MATLAB scripts.

### Cholesterol probe D4H-mCherry and plasma membrane probe PM-GFP

The cholesterol probe D4H-mCherry was a fusion protein of the minimal cholesterol-binding domain 4 of the bacterial toxin Perfringolysin O and the fluorescent protein mCherry (*Lim et al., 2019*; *Maekawa and Fairn, 2015*). The plasma membrane probe PM-GFP was constructed by inserting the 10 amino acid motif sufficient for palmitoylation and myristoylation modifications from the N-terminus of Lyn protein to the N-terminus of GFP. The PM-GFP was targeted to the plasma membrane by palmitoylation and myristoylation modifications (*Pyenta et al., 2001*). Oligonucleotides used in probe constructions are listed in the Key Resources Table Appendix.

### Immunofluorescent staining

Larvae were fixed in 4% paraformaldehyde/1 x PBS for 1 hr at room temperature and then overnight at 4 °C. Fixed samples were treated with methanol overnight at –20°C, and then step-washed in PBST

(1 x PBST/0.1% Tween-20). Fixed larvae were permeabilized in 1% Triton X-100/1 x PBS for 1 hr at room temperature. Cultured cells grown on coverslips were fixed with 4% paraformaldehyde/1 x PBS for 10 min at room temperature, and then post-fixed with methanol at –20°C for 15 min. Fixed cells were permeabilized with 0.5% Triton X-100/1 x PBS for 7 min. Fixed and permeabilized samples were blocked in PBS containing 2% sheep serum, 2% goat serum, 0.2% BSA, and 0.1% Tween-20. Blocked samples were washed, incubated with primary antibodies, and then with secondary antibodies.

## Microscopy and quantification of fluorescence signal

For laser scanning confocal microscopy, live or fixed larvae were mounted with 1.6% low-melting-temperature agarose and imaged on an Olympus confocal system with a 40 x water-immersion objective. For structured illumination microscopy, fixed larvae were cut into 20-μm-thick sections in a cryostat, then imaged on a Nikon N-SIM super-resolution microscope with a 100 x oil-immersion objective. Fluorescence signals were quantified with custom MATLAB scripts (Source_code.zip). In brief, raw confocal data were imported using Bio-Formats (*Linkert et al., 2010*), segmented with an adaptive thresholding method, then the selected pixels were quantified. For data plotted in *Figures 1D, E, 4D, E, 5A, B, C, E, 6A, B and C*, and *Figure 6—figure supplement 1A–B*, raw confocal data were imported into ImageJ using Bio-Formats, regions of the inner ear (30 μm x 60 μm x 10 slices, with 1.5 μm step/slice) or lateral line neuromast (30 μm x 30 μm x 10 slices, with 1.5 μm step/slice) hair cells were selected, then batch-processed and quantified with the confocalQuant.m script. For *Figure 3B* and *Figure 7B*, raw confocal data were directly read, batch-processed, and quantified with the cholesterolQuant.m scripts. For *Figure 4—figure supplement 1B*, raw confocal data were imported into ImageJ, stereocilia and basolateral regions were marked with polygon selection tool onto the hair cells expressing PM-GFP/D4H-mCherry, then the area and the gray values of the fluorescence were measured with ImageJ measure tool.

## Adult inner ear dissection and analysis

Dissections of inner ears in adult zebrafish were carried out following published protocols (*Liang and Burgess, 2009*; *Monroe et al., 2016*). The dissected inner ear sensory epithelia (utricle and saccule) were incubated with fluorescence-conjugated phalloidin (Beyotime, Shanghai) for 30 min at room temperature. For quantification, three images were acquired for each utricle (area 1: center of the extrastriolar region; area 2 and 3: lateral and medial striolar region, respectively), or each saccule (area1, 2, and 3: 5%, 50%, and 90% positions along the anterior-posterior axis). Hair bundles were counted within a 50x50 μm region centered on each image using the ImageJ multi-point tool.

## Paraffin sectioning and hematoxylin-eosin staining

Adult zebrafish were anesthetized and then fixed in 4% paraformaldehyde overnight at room temperature. Fixed animals were decalcified, dehydrated, and embedded in paraffin. The solidified block was trimmed and cut into 3-μm-thick sections. The sections (one per 10 sections) were collected between the utricle and saccule regions. Hematoxylin-eosin staining was carried out following standard protocols. Images were acquired on an Olympus VS120 slide scanning system with a 40 x objective.

## Immunoblot analysis

Cultured cells or zebrafish larvae were lysed in 2 x SDS sample buffer and then boiled at 95°C for 10 min to extract proteins. Extracted proteins were resolved by 6–20% SDS-PAGE, and then tank-transferred onto nitrocellulose membranes. Blotted membranes were probed with primary antibodies overnight and detected with HRP-conjugated secondary antibodies.

## Pharmacological treatment

2-Hydroxypropyl-β-cyclodextrin (2HPβCD) stock (200 mM) was prepared in EM. Larvae were treated with 7.5 mM 2HPβCD diluted in EM from 2 dpf to the designated time, and the treatment media were renewed every day. U18666A stock (10 mM) was prepared in DMSO. Larvae were treated with 7 μM U18666A diluted in EM from 2 dpf to the designated time, and the treatment media were renewed every day. Efavirenz stock (20 mM) and voriconazole stock (10 mg/ml) were prepared in DMSO. Larvae were treated with 2 μM efavirenz, or 1 μg/ml voriconazole diluted in EM from 2 dpf to the designated

time. For solvent controls, larvae were exposed to the same concentrations of EM or DMSO as in the drug-treated groups.

## AutoDock analysis of cholesterol interaction

Docking was performed using AutoDock 4.2 and AutoDockTools (ADT 1.5.7) as previously described (*Forli et al., 2016*; *Morris et al., 2009*). The PDB file for MINAR2 was downloaded from Alphafold (https://alphafold.com/entry/P59773). The ligands including cholesterol (CLR) and cholesterol analogs (desmosterol, MHQ; 25-hydroxy cholesterol, HC3; cholesterol-3-sulfate, C3S; estradiol, EST) were prepared by importing ideal coordinates from the Ligand-Expo of RCSB (http://ligand-expo.rcsb.org/ld-search.html).

## Statistical analysis

Sample sizes were estimated based on data variances and effect sizes in pilot experiments. Data analyses and plots were carried out using GraphPad Prism 9. Significance values are denoted as, $^*$: $p<0.05$, $^{**}$: $p<0.01$, $^{***}$: $p<0.001$, $^{****}$: $p<0.0001$. Each statistical test is run on data from at least three biological repeats. Error bars in line and column plots represent standard errors. Dashed white lines in violin plots represent medians and quartiles. Sample sizes for each figure are given in the figure legends. The significance of differences was assessed by two-tailed Student's *t*-test, one sample *t*-test, one-way ANOVA (ordinary or Brown-Forsythe), two-way ANOVA, or non-parametric analysis when appropriate.

## Acknowledgements

We thank Dr. Teresa Nicolson for providing the sequence information of the *myo6b* constructs. We thank Drs. Wei-Jun Pan, Qing-Jian Han, Ying Cao, and members of the laboratory for helpful discussions. We thank X Li and C Han for fish care. This work was supported by the National Key Research and Development Program of China (2018YFA0801000), National Natural Science Foundation of China (31571067), Shanghai Municipal Science and Technology Major Project (No.2018SHZDZX01), ZJ Lab, and Shanghai Center for Brain Science and Brain-Inspired Technology.

## Additional information

### Funding

| Funder | Grant reference number | Author |
|---|---|---|
| National Key Research and Development Program of China | 2018YFA0801000 | Gang Peng |
| National Natural Science Foundation of China | 31571067 | Gang Peng |
| Shanghai Municipal Science and Technology Commission | Major Project No.2018SHZDZX01 | Gang Peng |

The funders had no role in study design, data collection and interpretation, or the decision to submit the work for publication.

### Author contributions

Ge Gao, Data curation, Formal analysis, Validation, Investigation, Visualization, Methodology, Writing - original draft; Shuyu Guo, Data curation, Formal analysis, Investigation, Visualization, Writing - original draft; Quan Zhang, Hefei Zhang, Resources, Investigation, Methodology; Cuizhen Zhang, Resources, Project administration; Gang Peng, Conceptualization, Resources, Formal analysis, Supervision, Funding acquisition, Methodology, Project administration, Writing - review and editing

### Author ORCIDs

Gang Peng ⓘ http://orcid.org/0000-0001-6625-5426

### Ethics

The animal use protocols were approved by the Fudan University Shanghai Medical College Institution Animal Care and Use Committee (130227-092, 150119-088 and 190221-147). All animals were handled in accordance with the Fudan University Regulations on Animal Experiments.

### Decision letter and Author response

Decision letter https://doi.org/10.7554/eLife.80865.sa1
Author response https://doi.org/10.7554/eLife.80865.sa2

## Additional files

### Supplementary files

- MDAR checklist
- Source code 1. MATLAB scripts for quantification of fluorescence signals.

### Data availability

All data generated or analyzed during this study are included in the manuscript and supporting files. Source data files have been provided.

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

# Appendix 1

## Appendix 1—key resources table

| Reagent type (species) or resource | Designation | Source or reference | Identifiers | Additional information |
|---|---|---|---|---|
| Strain, strain background (*Danio rerio*) | AB | University of Oregon Zebrafish Facility | N/A | |
| Genetic reagent (*Danio rerio*) | minar2*fs139* | This paper | N/A | Available from G. Peng (the lead contact)'s lab |
| Genetic reagent (*Danio rerio*) | minar2*fs140* | This paper | N/A | Available from G. Peng (the lead contact)'s lab |
| Genetic reagent (*Danio rerio*) | Tg (UAS:EGFP) | *Asakawa et al., 2008* | N/A | |
| Genetic reagent (*Danio rerio*) | Tg (*myo6b*: GAL4FF) | This paper | N/A | Available from G. Peng (the lead contact)'s lab |
| Genetic reagent (*Danio rerio*) | Tg (*myo6b*: GFP-Minar2) | This paper | N/A | Available from G. Peng (the lead contact)'s lab |
| Genetic reagent (*Danio rerio*) | Tg (*myo6b*: Lamp1-GFP) | This paper | N/A | Available from G. Peng (the lead contact)'s lab |
| Genetic reagent (*Danio rerio*) | Tg (*myo6b*: Lamp1-mCherry) | This paper | N/A | Available from G. Peng (the lead contact)'s lab |
| Genetic reagent (*Danio rerio*) | Tg (*myo6b*: PM-GFP) | This paper | N/A | Available from G. Peng (the lead contact)'s lab |
| Genetic reagent (*Danio rerio*) | Tg (*myo6b*: D4H-mCherry) | This paper | N/A | Available from G. Peng (the lead contact)'s lab |
| Cell line (*Homo sapiens*) | HEK-293 | Chinese Academia of Sciences Cell Bank | SCSP-502 | |
| Cell line (*Homo sapiens*) | HeLa | Chinese Academia of Sciences Cell Bank | TCHu187 | |
| Cell line (*Cercopithecus aethiops*) | Cos-7 | Chinese Academia of Sciences Cell Bank | SCSP-508 | |
| Antibody | anti-Tubulin Acetylated antibody (Mouse monoclonal) | Sigma | T6793 | IF(1:1000) |
| Antibody | anti-GCC1 (Rabbit polyclonal) | Sigma | 021323 | IF(1:200) |
| Antibody | anti-GM130 (Mouse monoclonal) | BD | 610822 | IF(1:200) |
| Antibody | anti-GFP (Mouse monoclonal) | Proteintech | 66002 | WB(1:2000) |
| Antibody | anti-β-actin (Mouse monoclonal) | Proteintech | 60008 | WB(1:1000) |
| Antibody | anti-mCherry (Mouse monoclonal) | Abmart | M40012 | WB(1:2000) |
| Antibody | anti-GAPDH (Mouse monoclonal) | Abmart | M20006 | WB(1:1000) |
| Antibody | anti-FLAG M2 (Mouse monoclonal) | Sigma | F3165 | IF(1:1000) |
| Antibody | anti-FLAG M2 (Rabbit monoclonal) | Cell Signaling Technology | 14793 S | IF(1:500) |
| Antibody | Alexa Fluor 488 Goat anti-Mouse IgG (H+L) (Goat polyclonal) | Thermo Fisher Scientific | A11001 | IF(1:500) |
| Antibody | Alexa Fluor 488 Goat anti-Rabbit IgG (H+L) (Goat polyclonal) | Thermo Fisher Scientific | A11034 | IF(1:500) |
| Antibody | Alexa Fluor 546 Goat anti-Mouse IgG (H+L) (Goat polyclonal) | Thermo Fisher Scientific | A11003 | IF(1:500) |

*Appendix 1 Continued on next page*

*Appendix 1 Continued*

| Reagent type (species) or resource | Designation | Source or reference | Identifiers | Additional information |
|---|---|---|---|---|
| Antibody | Alexa Fluor 546 Goat anti-Rabbit IgG (H+L) (Goat polyclonal) | Thermo Fisher Scientific | A11035 | IF(1:500) |
| Antibody | Anti-mouse IgG, HRP-linked antibody (Horse polyclonal) | Cell Signaling Technology | 7076 | WB(1:8000) |
| Antibody | Anti-rabbit IgG, HRP-linked antibody (Goat polyclonal) | Cell Signaling Technology | 7074 | WB(1:8000) |
| Recombinant DNA reagent | pCS2-mCherry-MINAR2 | This paper | N/A | Available from G. Peng (the lead contact)'s lab |
| Recombinant DNA reagent | pCS2-EGFP-MINAR2 | This paper | N/A | Available from G. Peng (the lead contact)'s lab |
| Recombinant DNA reagent | pCS2-EGFP-MINAR2$^{YW-A}$ | This paper | N/A | Available from G. Peng (the lead contact)'s lab |
| Recombinant DNA reagent | Endoplasmic targeting KDEL | *Kneen et al., 1998*; *Sasavage et al., 1982* | N/A | |
| Recombinant DNA reagent | pCS2-EGFP-KDEL | This paper | N/A | Available from G. Peng (the lead contact)'s lab |
| Recombinant DNA reagent | Plasma membrane targeting PM | *Pyenta et al., 2001*; *Wu et al., 2004* | N/A | |
| Recombinant DNA reagent | pCS2-PM-EGFP | This paper | N/A | Available from G. Peng (the lead contact)'s lab |
| Recombinant DNA reagent | D4H | *Lim et al., 2019*; *Maekawa and Fairn, 2015* | N/A | |
| Recombinant DNA reagent | pCS2-D4H-mCherry | This paper | N/A | Available from G. Peng (the lead contact)'s lab |
| Recombinant DNA reagent | *myo6b*: D4H$^{T490G-L491G}$-mCherry | This paper | N/A | Available from G. Peng (the lead contact)'s lab |
| Recombinant DNA reagent | *myo6b*:GFP-P2A-FLAG-Minar2 | This paper | N/A | Available from G. Peng (the lead contact)'s lab |
| Recombinant DNA reagent | *myosin 6b* gene promoter | *Kindt et al., 2012* | N/A | |
| Chemical compound, drug | 2-hydroxypropyl-β-cyclodextrin (2HPβCD) | Sangon Biotech | A600388 | |
| Chemical compound, drug | AM1-43 | Biotium | 70024 | |
| Chemical compound, drug | Efavirenz | MCE | HY-10572 | |
| Chemical compound, drug | Filipin | MCE | HY-N6716 | |
| Chemical compound, drug | Lyso-Tracker Red | Beyotime | C1046 | |
| Chemical compound, drug | Phalloidin | Beyotime | C1033/C2203S | |
| Chemical compound, drug | U18666A | MCE | HY-107433 | |
| Chemical compound, drug | Voriconazole | MCE | HY-76200 | |
| Software, algorithm | MATLAB | Mathworks | https://www.mathworks.com/products/matlab.html | |
| Software, algorithm | GraphPad Prism 9 | GraphPad Software | https://www.graphpad.com/scientificsoftware/prism/ | |
| Software, algorithm | ImageJ | *Schneider et al., 2012* | https://imagej.nih.gov/ij/index.html | |
| Software, algorithm | Clustal Omega sequence alignment software | European Bioinformatics Institute | https://www.ebi.ac.uk/Tools/msa/clustalo/ | |
| Software, algorithm | FigTree v1.4.2 | Andrew Rambaut | http://tree.bio.ed.ac.uk/software/figtree/ | |
| Software, algorithm | GIMP 2.8.14 | GNU Image Manipulation Program | https://www.gimp.org/downloads/ | |

*Appendix 1 Continued on next page*

*Appendix 1 Continued*

| Reagent type (species) or resource | Designation | Source or reference | Identifiers | Additional information |
|---|---|---|---|---|
| Software, algorithm | GeneDoc | NRBSC | http://nrbsc.org/gfx/genedoc | |
| Software, algorithm | Flycapture2 | Teledyne FLIR | https://www.flir.com/products/flycapture-sdk/ | |
| Software, algorithm | FV10-ASW 4.2 Viewer | Olympus | https://www.olympus-lifescience.com.cn/en/ | |
| Sequence-based reagent | *minar2*^fs139 WT genotyping_F | This paper | PCR primers | TGGGAATGTTGCCGGCTACACAT |
| Sequence-based reagent | *minar2*^fs139 WT genotyping_R | This paper | PCR primers | AGCCTACTATTGTAGTTGTATTACC |
| Sequence-based reagent | *minar2*^fs139 HO genotyping_F | This paper | PCR primers | GGAATGTTGCCGGCATGGAA |
| Sequence-based reagent | *minar2*^fs139 HO genotyping_R | This paper | PCR primers | AGCCTACTATTGTAGTTGTATTACC |
| Sequence-based reagent | *minar2*^fs140 WT genotyping_F | This paper | PCR primers | GTTGCCGGCTACACATGGAA |
| Sequence-based reagent | *minar2*^fs140 WT genotyping_R | This paper | PCR primers | AGCCTACTATTGTAGTTGTATTACC |
| Sequence-based reagent | *minar2*^fs140 HO genotyping_F | This paper | PCR primers | GAATGTTGCCGGCTACATGGAAC |
| Sequence-based reagent | *minar2*^fs140 HO genotyping_R | This paper | PCR primers | AGCCTACTATTGTAGTTGTATTACC |
| Sequence-based reagent | *minar2* target site verification_F | This paper | PCR primers | AGTAGGTATCAGGTAGAGTTACAT |
| Sequence-based reagent | *minar2* target site verification_R | This paper | PCR primers | AGCCTACTATTGTAGTTGTATTACC |
| Sequence-based reagent | *zebrafish-minar2_F* | This paper | RT-PCR primers | CAACGGCAGTGGCACAACAGGAT |
| Sequence-based reagent | *zebrafish-minar2_R* | This paper | RT-PCR primers | GTGAAGTGTGTCTGTCATAGTCCTG |
| Sequence-based reagent | *zebrafish-minar1a_F* | This paper | RT-PCR primers | CAGGTCCAGGAATCACTCAACC |
| Sequence-based reagent | *zebrafish-minar1a_R* | This paper | RT-PCR primers | GCGGGGAAAAAATAAAGATAGAAACC |
| Sequence-based reagent | *zebrafish-minar1b_F* | This paper | RT-PCR primers | CCAGGAGCCACACAGAGAGC |
| Sequence-based reagent | *zebrafish-minar1b_R* | This paper | RT-PCR primers | CGGTGTGTAAATCTCATCTGTCCA |
| Sequence-based reagent | *zebrafish-gapdh_F* | This paper | RT-PCR primers | CATCGTTGAAGGTCTTATGAGCACTG |
| Sequence-based reagent | *zebrafish-gapdh_R* | This paper | RT-PCR primers | AGGTTTCTCAAGACGGACTGTCAG |
| Sequence-based reagent | *zebrafish-hmgcra_F* | This paper | RT-PCR primers | GATTGAGCCTGACATGCCCCTG |
| Sequence-based reagent | *zebrafish-hmgcra_R* | This paper | RT-PCR primers | GCAGGGGTCGAATCACTAAATCTC |
| Sequence-based reagent | *zebrafish-hmgcs1_F* | This paper | RT-PCR primers | ATGGGATTCTGCTCGGACCGC |
| Sequence-based reagent | *zebrafish-hmgcs1_R* | This paper | RT-PCR primers | CATACACAGCAATATCACCAGCAAC |
| Sequence-based reagent | *zebrafish-mylipa_F* | This paper | RT-PCR primers | GAATCTCCCAGCAGATGGACAATC |
| Sequence-based reagent | *zebrafish-mylipa_R* | This paper | RT-PCR primers | TGTGCTTGGCTATGATACTGTTGATG |
| Sequence-based reagent | *zebrafish-abcg1_F* | This paper | RT-PCR primers | GCCCTGGAGCTGGTCAACAAC |

*Appendix 1 Continued on next page*

*Appendix 1 Continued*

| Reagent type (species) or resource | Designation | Source or reference | Identifiers | Additional information |
|---|---|---|---|---|
| Sequence-based reagent | *zebrafish-abcg1_R* | This paper | RT-PCR primers | TATTCACCAGACG CCACCTCCATT |
| Sequence-based reagent | *zebrafish probe minar2_F* | This paper | PCR primers | AGTCACAAAATGG ACATAGCCGTC |
| Sequence-based reagent | *zebrafish probe minar2_R* | This paper | PCR primers | CTAGATTGTAGAG CAGGGTTGTTC |
| Sequence-based reagent | *myo6b* promoter section1_F | This paper | PCR primers | ccagtttaatttGTACACCTGT CCAACTGCTCATTAG |
| Sequence-based reagent | *myo6b* promoter section1_R | This paper | PCR primers | cAAGTCACAAGGTGC CTACTGGGTTGCC |
| Sequence-based reagent | *myo6b* promoter section2_F | This paper | PCR primers | ggcaccttgtgacttAACCCAG TAGGCACCTTGTGACTT |
| Sequence-based reagent | *myo6b* promoter section2_R | This paper | PCR primers | ccccaTTATTTACAGT GTAAAATTCTTTG |
| Sequence-based reagent | *myo6b* promoter section3_F | This paper | PCR primers | ctgtaaataaTGGGGTCG CCACAGCGGAATGAAC |
| Sequence-based reagent | *myo6b* promoter section3_R | This paper | PCR primers | ataagtacgggatctATTG CACCCCACAATT ACTCCACAGCTCTG |
| Sequence-based reagent | p-mTol2-*myo6b*:GAL4FF_F | This paper | PCR primers | tggggtgcaatAAATAGAT CCCGTACTTATATAAG |
| Sequence-based reagent | p-mTol2-*myo6b*:GAL4FF_R | This paper | PCR primers | ggacaggtgtacAAATTA AACTGGGCATCAGCGC |
| Sequence-based reagent | zebrafish *minar2* cDNA_F | This paper | RT-PCR primers | ATGGACATAGCCG TCCTGCCGAAC |
| Sequence-based reagent | zebrafish *minar2* cDNA_R | This paper | RT-PCR primers | TCAGTCTCTTGATT GTTTTACCACTAT |
| Sequence-based reagent | *myo6b*:GFP-Minar2_F | This paper | PCR primers | aaatagatcccATGGTGA GCAAGGGCGAGGAG |
| Sequence-based reagent | *myo6b*:GFP-Minar2_R | This paper | PCR primers | gattagttacccTCAGTC TCTTGATTGTTTTACC |
| Sequence-based reagent | p-mTol2-*myo6b*:GFP-Minar2_F | This paper | PCR primers | tcaagagactgaGGGTA ACTAATCTAGAACTATAG |
| Sequence-based reagent | p-mTol2-*myo6b*:GFP-Minar2_R | This paper | PCR primers | cttgctcaccatGGGATCT ATTTATTGCACCCCA |
| Sequence-based reagent | p-mTol2-*myo6b*:GFP-P2A-FLAG-Minar2 (P2A section)_F | This paper | PCR primers | gagctgtacaagGGAAGC GGAGCTACTAACTTC |
| Sequence-based reagent | p-mTol2-*myo6b*:GFP-P2A-FLAG-Minar2 (P2A section)_R | This paper | PCR primers | cggatcctgcaaAGGTCC AGGGTTCTCCTCC |
| Sequence-based reagent | p-mTol2-*myo6b*:GFP-P2A-FLAG-Minar2 (FLAG section)_F | This paper | PCR primers | aaccctggacctTTGCAG GATCCGATGGACTAC |
| Sequence-based reagent | p-mTol2-*myo6b*:GFP-P2A-FLAG-Minar2 (FLAG section)_R | This paper | PCR primers | ggctatgtccatTCCAGAA CCTTTGTCATCGTC |
| Sequence-based reagent | p-mTol2-*myo6b*:GFP-P2A-FLAG-Minar2 (Minar2 section)_F | This paper | PCR primers | aaaggttctggaATGGACAT AGCCGTCCTGC |
| Sequence-based reagent | p-mTol2-*myo6b*:GFP-P2A-FLAG-Minar2 (Minar2 section)_R | This paper | PCR primers | agctccgcttccCTTGTAC AGCTCGTCCATGC |
| Sequence-based reagent | D4H_F | This paper | PCR primers | ATGAAGGGAAAA ATAAACTTAGATC |
| Sequence-based reagent | D4H_R | This paper | PCR primers | ATTGTAAGTAAT ACTAGATCCAGG |

*Appendix 1 Continued on next page*

*Appendix 1 Continued*

| Reagent type (species) or resource | Designation | Source or reference | Identifiers | Additional information |
|---|---|---|---|---|
| Sequence-based reagent | *myo6b*:D4H-mCherry_F | This paper | PCR primers | taaatagatcccgccaccA TGAAGGGAAAAATAAA |
| Sequence-based reagent | *myo6b*:D4H-mCherry_R | This paper | PCR primers | gattagttacccTTACTTG TACAGCTCGTCCATG |
| Sequence-based reagent | p-mTol2-*myo6b*:D4H-mCherry_F | This paper | PCR primers | ctgtacaagtaaGGGTAA CTAATCTAGAACTATAG |
| Sequence-based reagent | p-mTol2-*myo6b*:D4H-mCherry_R | This paper | PCR primers | ttcccttcatggtggcGGG ATCTATTTATTG |
| Sequence-based reagent | p-mTol2-*myo6b*:D4H$^{T490G-L491G}$-mCherry_F | This paper | PCR primers | tggggaacaggcGGATACC CTGGATCTAGTATTAC |
| Sequence-based reagent | p-mTol2-*myo6b*:D4H$^{T490G-L491G}$-mCherry_R | This paper | PCR primers | tccagggtatccGCCTGTT CCCCATATTGAAACAT |
| Sequence-based reagent | *myo6b*:PM-GFP_F | This paper | PCR primers | gatcccGCCACCatgggttgt aaaaaatccaagttggatggtgacc aaaatggatgtgtgcttgaaccagt gaacGGTTCTGGAATG |
| Sequence-based reagent | *myo6b*:PM-GFP_R | This paper | PCR primers | CATTCCAGAACCgttc actggttcaagcacacatccattttggt caccatccaacttggatttttta caacccatGGTGGCgggatc |
| Sequence-based reagent | p-mTol2-*myo6b*:PM-GFP_F | This paper | PCR primers | gaaccagtgaacGGTTCT GGAATGGTGAGCAAGG |
| Sequence-based reagent | p-mTol2-*myo6b*:PM-GFP_R | This paper | PCR primers | tttacaacccatGGTGGCGG GATCTATTTATTGCAC |
| Sequence-based reagent | zebrafish *lamp1*_F | This paper | RT-PCR primers | ATGGCGCGAG CTGCAGGTGT |
| Sequence-based reagent | zebrafish *lamp1*_R | This paper | RT-PCR primers | GATGGTCTGGT ACCCGGCGT |
| Sequence-based reagent | *myo6b*:Lamp1-GFP/mCherry_F | This paper | PCR primers | taccagaccatcGGTTC TGGAATGGTGAGCAAGG |
| Sequence-based reagent | *myo6b*:Lamp1-GFP/mCherry_R | This paper | PCR primers | agctcgcgccatGGTGG CGGGATCTATTTATTGCAC |
| Sequence-based reagent | p-mTol2-*myo6b*:Lamp1-GFP/mCherry_F | This paper | PCR primers | gatcccgccaccATGGC GCGAGCTGCAGGTGT |
| Sequence-based reagent | p-mTol2-*myo6b*:Lamp1-GFP/mCherry_R | This paper | PCR primers | ccattccagaaccGATGG TCTGGTACCCGGCGT |
| Sequence-based reagent | human *MINAR2* cDNA_F | This paper | RT-PCR primers | aattgccaccATGGATCTC TCTGTTTTGCCAAATAACAA |
| Sequence-based reagent | human *MINAR2* cDNA_R | This paper | RT-PCR primers | ccggGGTGAAAAAAG TAATGATAGTCACTATGG |
| Sequence-based reagent | pCS2-eGFP-MINAR2$^{1-190}$_F | This paper | PCR primers | acTGTACAAGat ggatctctctgttttgcc |
| Sequence-based reagent | pCS2-eGFP-MINAR2$^{1-190}$_R | This paper | PCR primers | gcagCGAGCTCTTA ggtgaaaaaagtaatgatagtc |
| Sequence-based reagent | pCS2-eGFP-MINAR2$^{YW-A}$ step1_F | This paper | PCR primers | GCTACCATTGAGGA AGCAGACAAACATT CCCTGCACACA |
| Sequence-based reagent | pCS2-eGFP-MINAR2$^{YW-A}$ step1_R | This paper | PCR primers | AACTTAGGTCACCTG CGAGTGGGTTATT CTTCATAACTG |
| Sequence-based reagent | pCS2-eGFP-MINAR2$^{YW-A}$ step2_F | This paper | PCR primers | GCTATGGAAGAAAG AAAAAGAACCCCTC AGCTACCATTGAG GAAGCAGAC |
| Sequence-based reagent | pCS2-eGFP-MINAR2$^{YW-A}$ step2_R | This paper | PCR primers | TCTTTTTTCTTTCTTC CATAGCTTCCTCC AAACTTAGGTCACC TGCGAGTG |

