## [Editor Report]

This is an important study linking cholesterol homeostasis to sensory hair cell function. Using the knockout approach in zebrafish, the authors provided compelling evidence that Minra2, a gene known to be associated with deafness in humans, encodes a protein that functions to regulate cholesterol homeostasis in the hair bundle of sensory hair cells.

---

## [Decision Letter]

**Decision letter after peer review:**

Thank you for submitting your article "Kiaa1024L/Minar2 is essential for hearing by regulating cholesterol distribution in hair bundles" for consideration by *eLife*. Your article has been reviewed by 3 peer reviewers, and the evaluation has been overseen by a Reviewing Editor and Kathryn Cheah as the Senior Editor. The following individual involved in the review of your submission has agreed to reveal their identity: David W Raible (Reviewer #2).

Essential revisions:

All three reviewers are impressed with the finding of Minar2's involvement in regulating cholesterol and hair cell function. Nevertheless, there are some concerns. Adding a control of using a non-binding mutant version of D4H as suggested by reviewer #1 is important. Second, questions were raised about statements such as the accessible pool of cholesterol being enriched, yet no data on the assessment of other cholesterol pools was provided. If this cannot be addressed experimentally in a timely manner, then tone down the statements in the text.

*Reviewer #1 (Recommendations for the authors):*

1. All of the accessible cholesterol measurements are carried out using an intracellularly expressed version of D4H. To eliminate complications due to aggregation of the probe when not bound, the authors need to confirm the fidelity of their results by using a non-binding mutant version of D4H.

2. Since the authors implicate that it is not total cholesterol, but a sub-pool of cholesterol that is important for hearing, it would be important to look at other pools of cholesterol to establish this point – there are many reported probes that monitor phospholipid-bound cholesterol.

3. The in vitro interaction docking studies of Minar2 with cholesterol could be more informative if docking was carried out with several structural analogs of cholesterol to get at specificity.

*Reviewer #2 (Recommendations for the authors):*

The authors need to address the following points:

1. Authors should include more details in their protocol section. Several experiments lack a proper description in the methods section, like methods for Figure 1-Supp2-C which are not mentioned at all, or others like Figure 4C, where myo6b:PM-GFP is mentioned in the main text, but not in the methods. This makes it challenging to understand some of the figures. Moreover, the authors rely heavily on image analysis throughout the paper, and only a small paragraph with no details is presented. If the authors developed custom analysis pipelines in MATLAB, these should be described in detail in the methods section or included as supplemental. Lastly, there should be a section describing microscopy acquisition techniques and settings, for both non- and super-resolution images.

2. Image quality could be better, with higher magnification images in some cases. Figure 4A is not very convincing, and authors could get a better image, as well as include a neuromast image. Figure legends are well described, but authors should also be consistent in the details given within these as some information is missing (i.e. number of fish used on Figure 1 Supp.2-C, or replicates performed).

3. Additional alternatives for the differences between zebrafish and mice should be considered. The very recent PNAS paper that came out after submission (PMID: 35727972) shows that there is a significant loss as early as two weeks in mice. Potential alternatives include compensation by other gene family members due to mutant mRNA degradation (PMID: 30944477).

4. The authors should test whether cholesterol-lowering drugs (Figure 5) result in a decrease in hair cells in both wt and mutant. In addition, these studies would be strengthened if startle response was tested as for the Cyp inhibitors.

5. The authors mention two alleles but all assays are with only one. It would be worth checking that important results are replicable in both (e.g. startle response rescue by Efa).

6. There is some mismatch between text and figures, e.g. Figure 3B (line 429) and Figure 6A (line 658).

7. Co-localization of lysotracker and GFP-Minar2 should also be tested in HEK-293 (Figure 2-FS1), as these are the cells used for later analysis (Figure 7). Which cells are used in Figure 2-FS1 panel F? Images of control for that panel are needed for comparison.

8. Discussion of Minar1 and Sars-CoV2 are largely a tangent and could be deleted or extensively reduced.

9. The authors should make figures color-blind accessible by the use of magenta instead of red.

*Reviewer #3 (Recommendations for the authors):*

1. Can the authors validate the localization of MINAR2 in another way? Maybe by labeling mouse hair cells with an appropriate antibody or by expression of the MINAR2 with a small epitope tag, not a fluorescent protein in zebrafish hair cells. This would independently validate.

2. As stated above, can the authors perform EM? Also, are there defects in the neuromasts in mutants?

3. Figure 1A do prim 1- and prim 2-derived neuromasts express minar2? Do anterior neuromasts express minar2?

4. The author should figure out a way to determine if the PM of stereocilia has higher levels of cholesterol than that of the basolateral PM. They need to take into account that there is a much higher level of PM in stereocilia.

5. Perform a genomics search and let us know if there are 2 copies of Kiaa1024L/Minar2 in the zebrafish genome.

---

## [Author Response]

Essential revisions:All three reviewers are impressed with the finding of Minar2's involvement in regulating cholesterol and hair cell function. Nevertheless, there are some concerns. Adding a control of using a non-binding mutant version of D4H as suggested by reviewer #1 is important.

The reviewer raised an important question, and we thank the reviewers for their suggestions. We have examined a non-binding D4H^T490G-L491G^ mutant and found it gave rise to diffused signals in the cytoplasm of hair cells (revised Figure 4B). This finding indicated that the wild-type D4H probe's stereocilia localization is dependent on its ability to bind to cholesterol. We have added this information to the revised manuscript:

Line 364-372

“To exclude the complications that the D4H-mCherry probe may aggregate in the stereocilium region without binding to cholesterol, we constructed a non-binding D4H^T490G-L491G^ variant. Previous studies have shown that the threonine–leucine pair in loop L1 of PFO domain 4 is essential for cholesterol recognition, and substitution of the T490-L491 pair with glycine residues abolishes cholesterol binding without perturbation of the overall structure of PFO (Farrand et al., 2010). When the D4H^T490G-L491G^-mCherry variant was expressed in the hair cells, it gave rise to diffused signals in the cytoplasm only (Figure 4B)”

Second, questions were raised about statements such as the accessible pool of cholesterol being enriched, yet no data on the assessment of other cholesterol pools was provided. If this cannot be addressed experimentally in a timely manner, then tone down the statements in the text.

We agree that it is not yet known if other cholesterol pools are also implicated. Since we haven't had experience with other cholesterol probes, we couldn't address this question in a timely manner. We have followed the suggestion, revised the text, and toned down our statements about the accessible cholesterol pool throughout the manuscript. We will develop the required probes and tools to examine other cholesterol pools in the hair cells.

Reviewer #1 (Recommendations for the authors):1. All of the accessible cholesterol measurements are carried out using an intracellularly expressed version of D4H. To eliminate complications due to aggregation of the probe when not bound, the authors need to confirm the fidelity of their results by using a non-binding mutant version of D4H.

We have used a non-binding mutant version of D4H as the reviewer suggested. We constructed a non-binding D4H^T490G-L491G^ variant and expressed the D4H^T490G-L491G^-mCherry fusion construct in hair cells. The results showed that the non-binding D4H^T490G-L491G^ variant gave rise to diffused signals in the cytoplasm of hair cells (revised Figure 4B), indicating that the wild-type D4H probe's stereocilia localization is dependent on its ability to bind to cholesterol. We have added this information to the revised manuscript:

Line 364-372

“To exclude the complications that the D4H-mCherry probe may aggregate in the stereocilium region without binding to cholesterol, we constructed a non-binding D4H^T490G-L491G^ variant. Previous studies have shown that the threonine–leucine pair in loop L1 of PFO domain 4 is essential for cholesterol recognition, and substitution of the T490-L491 pair with glycine residues abolishes cholesterol binding without perturbation of the overall structure of PFO (Farrand et al., 2010). When the D4H^T490G-L491G^-mCherry variant was expressed in the hair cells, it gave rise to diffused signals in the cytoplasm only (Figure 4B)”

2. Since the authors implicate that it is not total cholesterol, but a sub-pool of cholesterol that is important for hearing, it would be important to look at other pools of cholesterol to establish this point – there are many reported probes that monitor phospholipid-bound cholesterol.

We agree with the reviewer that it is not yet known if other pools of cholesterol are also involved. We haven’t had experience with probes that monitor phospholipid-bound cholesterol to address this question in a timely manner, and we will develop the necessary tools to examine other cholesterol pools in the hair cells. We have followed the revision suggestion and revised the manuscript to tone down our statements about the accessible pool of cholesterol.

3. The in vitro interaction docking studies of Minar2 with cholesterol could be more informative if docking was carried out with several structural analogs of cholesterol to get at specificity.

We have conducted docking studies with cholesterol and cholesterol analogs as the reviewer suggested. The cholesterol analogs included desmosterol, 25-hydroxy cholesterol, cholesterol-3-sulfate, and estradiol. The results showed that most cholesterol analogs interacted with MINAR2 in the same way that cholesterol did, except for cholesterol-3-sulfate. We have added this information to the revised manuscript:

Line 542-551

“We next carried out docking studies with cholesterol analogs including desmosterol (MHQ, a precursor of cholesterol), 25-hydroxy cholesterol (HC3, an oxysterol), cholesterol-3-sulfate (C3S, a head modification of cholesterol), and estradiol (EST, a steroid). Except for cholesterol-3-sulfate, these cholesterol analogs interacted with MINAR2 in the same way that cholesterol did. The binding free energy was comparable for 25-hydroxy cholesterol (-8.0 kJ/mol, versus -8.1 KJ/mol for cholesterol), but was reduced for desmosterol (-6.5 kJ/mol) and estradiol (-7.4 kJ/mol). When compared to other cholesterol analogs, the docked orientation of cholesterol-3-sulfate was flipped upside down, and the binding free energy was mostly reduced (-6.1 kJ/mol).”

Reviewer #2 (Recommendations for the authors):The authors need to address the following points:1. Authors should include more details in their protocol section. Several experiments lack a proper description in the methods section, like methods for Figure 1-Supp2-C which are not mentioned at all, or others like Figure 4C, where myo6b:PM-GFP is mentioned in the main text, but not in the methods. This makes it challenging to understand some of the figures. Moreover, the authors rely heavily on image analysis throughout the paper, and only a small paragraph with no details is presented. If the authors developed custom analysis pipelines in MATLAB, these should be described in detail in the methods section or included as supplemental. Lastly, there should be a section describing microscopy acquisition techniques and settings, for both non- and super-resolution images.

We thank the reviewer for the valuable comments. We have added these details in the revised manuscript:

(1) Methods for Figure 1-Supp2-C

We added a section titled “Paraffin sectioning and hematoxylin-eosin staining” in the Methods.

Line: 846-852

“Paraffin sectioning and hematoxylin-eosin staining

Adult zebrafish were anesthetized and then fixed in 4% paraformaldehyde overnight at room temperature. Fixed animals were decalcified, dehydrated, and embedded in paraffin. The solidified block was trimmed and cut into 3 µm thick sections. The sections (one per 10 sections) were collected between the utricle and saccule regions. Hematoxylin-eosin staining was carried out following standard protocols. Images were acquired on an Olympus VS120 slide scanning system with a 40x objective.”

(2) Methods for myo6b:PM-GFP

We added a description of *myo6b*:PM-GFP in the section titled “Cholesterol probe D4H-mCherry and plasma membrane probe PM-GFP”.

Line: 794-802

“Cholesterol probe D4H-mCherry and plasma membrane probe PM-GFP

The cholesterol probe D4H-mCherry was a fusion protein of the minimal cholesterol-binding domain 4 of the bacteria toxin Perfringolysin O and the fluorescent protein mCherry (Lim et al., 2019; Maekawa and Fairn, 2015). The plasma membrane probe PM-GFP was constructed by inserting the 10 amino acid motif sufficient for palmitoylation and myristoylation modification from the N-terminus of Lyn protein to the N-terminus of GFP. The PM-GFP was targeted to the plasma membrane by the palmitoylation and myristoylation modification (Pyenta et al., 2001). Oligonucleotides used in probe constructions are listed in the Key Resources Table.”

(3) Image analysis with MATLAB

We included the MATLAB scripts used in our image analyses in the “Source code 1” file. We added a detailed description of image analysis in the section titled “Microscopy and quantification of fluorescence signal” in the Methods.

Line: 816-834

“Microscopy and quantification of fluorescence signal

For laser scanning confocal microscopy, live or fixed larvae were mounted with 1.6% low-melting-temperature agarose and imaged on an Olympus confocal system with a 40x water-immersion objective. For structured illumination microscopy, fixed larvae were cut into 20 μm thick sections in a cryostat, then imaged on a Nikon N-SIM super-resolution microscope with a 100x oil-immersion objective. Fluorescence signals were quantified with custom MATLAB scripts (Source_code.zip). In brief, raw confocal data were imported using Bio-Formats (Linkert et al., 2010), segmented with an adaptive thresholding method, then the selected pixels were quantified. For data plotted in Figure 1D, 1E, 4D, 4E, 5A, 5B, 5C, 5E, 6A, 6B, 6C, and Figure 6—figure supplement 1A-1B, raw confocal data were imported into ImageJ using Bio-Formats, regions of the inner ear (30 μm x 60 μm x 10 slices, with 1.5 μm step/slice) or lateral line neuromast (30 μm x 30 μm x 10 slices, with 1.5 μm step/slice) hair cells were selected, then batch-processed and quantified with the confocalQuant.m script. For Figure 3B and Figure 7B, raw confocal data were directly read, batch-processed, and quantified with the cholesterolQuant.m scripts. For Figure 4—figure supplement 1B, raw confocal data were imported into ImageJ, stereocilia and basolateral regions were marked with polygon selection tool onto the hair cells expressing PM-GFP/D4H-mCherry, then the area and the gray values of the fluorescence were measured with ImageJ measure tool.”

(4) Microscopy acquisition

We added a description of confocal microscopy and structured illumination microscopy in the section “Microscopy and quantification of fluorescence signal” in the Methods (see above).

2. Image quality could be better, with higher magnification images in some cases. Figure 4A is not very convincing, and authors could get a better image, as well as include a neuromast image. Figure legends are well described, but authors should also be consistent in the details given within these as some information is missing (i.e. number of fish used on Figure 1 Supp.2-C, or replicates performed).

We have revised the figures and text to address these comments. We have provided high-resolution TIFF for all figures in the revised manuscript.

(1) Related to Figure 4A

Image panels in Figure 4A were enlarged to better show the contents. The lower panel in Figure 4A shows a neuromast from the lateral line.

In addition, in response to Reviewer #3’s comments on data related to Figure 4A, we analyzed and compared the distribution of PM-GFP with that of D4H-mCherry in individual inner ear hair cells. The quantification results supported that there was an enrichment of D4H-mCherry in the stereocilium membrane over the basolateral membranes (Figure 4—figure supplement 1B, and line: 349-364). Please see the detailed analysis under Question #4 from Reviewer #3 below.

(2) Related to replicates for Figure 1 Supp.2C

We performed paraffin sectioning and hematoxylin-eosin staining twice, using fish from two batches of stocks. Each replicate contained 1 wild type and 1 mutant fish. Data in Figure 1 Supp.2C were from one of the two replicates. Data from another replicate showed a similar reduction of hair cell numbers in the *minar2* mutant, but in this replicate the anterior utricle region was not collected during the paraffin sectioning (raw counts are provided in the source data file, Figure 1-source data 1.xlsx). We added the sample size and replicate information to the figure legend (Lines 1567-1569, “Results were from 1 wild type and 1 mutant fish. Sections of another wild type and mutant from a different batch of fish in another replicate showed a similar reduction of hair cells in the *minar2^fs139^* mutant.”).

Because we found it was impractical to rigorously quantify the number of hair cells with paraffin sectioning, we dissected utricles and saccules from inner ears and performed the quantification using whole-mount phalloidin staining of hair cells (Figure 1F).

3. Additional alternatives for the differences between zebrafish and mice should be considered. The very recent PNAS paper that came out after submission (PMID: 35727972) shows that there is a significant loss as early as two weeks in mice. Potential alternatives include compensation by other gene family members due to mutant mRNA degradation (PMID: 30944477).

We thank the reviewer for pointing this out. To assess if genetic compensation was involved in the less severe phenotype observed in the zebrafish, we had previously examined the expression levels of the UPF0258 gene family members in the zebrafish, which include two *minar1* orthologs (*ubtora/minar1a* and *ubtorb/minar1b*) and a single *minar2* gene. We found that the expression levels of *minar1a* and *minar1b* were not significantly changed in the *minar2* mutants (revised Figure 1—figure supplement 1E). We however can't exclude the possibility that changes in other genes outside the UPF0258 gene family compensate for the loss of *minar2*. We have revised the Discussion to include the alternatives of genetic compensation.

Line: 588-593

“Additional alternatives that may account for the less severe phenotype in the zebrafish include genetic compensation (El-Brolosy et al., 2019; Ma et al., 2019). Although the expression levels of *minar1a* and *minar1b* were not changed in mutant *minar2^fs139^* larvae, changes in other genes outside the UPF0258 gene family could partly remediate the loss of *minar2* in the zebrafish.”

4. The authors should test whether cholesterol-lowering drugs (Figure 5) result in a decrease in hair cells in both wt and mutant. In addition, these studies would be strengthened if startle response was tested as for the Cyp inhibitors.

We have performed these experiments as the reviewer suggested. The results showed that the cholesterol-lowering drugs 2HPβCD and U18666A both caused small reductions in hair cell numbers in wild-type controls, but only U18666A further reduced the numbers of hair cells in the neuromasts of mutant *minar2^fs139^* larvae (revised Figure 5—figure supplement 1B, and lines 478-482). Results of short-latency C-start response showed that 2HPβCD treatment greatly compromised startle response in both the wild type and the *minar2^fs139^* mutants (revised Figure 5—figure supplement 1A, and lines 455-458). Because treatment with U18666A caused overall defects including bend in the body, we didn’t perform C-start response tests for the U18666A treatment.

5. The authors mention two alleles but all assays are with only one. It would be worth checking that important results are replicable in both (e.g. startle response rescue by Efa).

Because we have archived the other allele (the *minar2^fs140^* allele), we couldn't conduct the startle response experiments in a timely manner. The two alleles, *minar2^fs139^* and *minar2^fs140^* were generated using the same targeting gRNA, and they both are frameshift mutants. We carried out initial experiments with both alleles, and the results were similar between these two alleles. Thus, we subsequently focused our investigation using the *minar2^fs139^* allele. The results with both alleles were presented in Figure 1—figure supplementary 1D (reduced mRNA levels, likely due to non-sense mediated decay), Figure 1—figure supplementary 1E (expression levels of *minar1a* and *minar1b*), Figure 1—figure supplementary 2A (number of hair cells), Figure 1—figure supplementary 2B (body length and weight), Figure 2C (length of stereocilium), Figure 2—figure supplement 1D (morphology of kinocilia bundle).

6. There is some mismatch between text and figures, e.g. Figure 3B (line 429) and Figure 6A (line 658).

We have corrected these mistakes and other mistakes we found in the manuscript.

7. Co-localization of lysotracker and GFP-Minar2 should also be tested in HEK-293 (Fig 2-FS1), as these are the cells used for later analysis (Fig 7). Which cells are used in Fig 2-FS1 panel F? Images of control for that panel are needed for comparison.

We have added a figure showing the co-localization of lysotracker and GFP-MINAR2 in the HEK293 cells (revised Figure 2-figure supplement 2C). The cells used in Fig 2-FS1 panel F were Cos-7 cells. We have also added the DMSO-treated control panel to this figure (revised Figure 2-figure supplement 2D).

8. Discussion of Minar1 and Sars-CoV2 are largely a tangent and could be deleted or extensively reduced.

We have removed the discussion of SARS-CoV-2 in the revised manuscript.

9. The authors should make figures color-blind accessible by the use of magenta instead of red.

We have revised the figures and replaced red with magenta for images showing both the red and green channels. We also checked all figures with the “Simulate Color Blindness” tool in ImageJ to ensure accessibility.

Reviewer #3 (Recommendations for the authors):1. Can the authors validate the localization of MINAR2 in another way? Maybe by labeling mouse hair cells with an appropriate antibody or by expression of the MINAR2 with a small epitope tag, not a fluorescent protein in zebrafish hair cells. This would independently validate.

We thank the reviewer for the valuable suggestion. We made a small epitope-tagged Minar2 construct as suggested. To facilitate the identification of hair cells and ensure the localization of Minar2 in hair cells, we made a *myo6b*:GFP-P2A-FLAG-Minar2 construct. The P2A self-cleaving products of the *myo6b*:GFP-P2A-FLAG-Minar2 construct allowed the labeling of hair cells by the GFP expression and localization of Minar2 by the small FLAG epitope tag. The results indicated that FLAG-Minar2 was similarly localized to the stereocilia and the apical region of hair cells (revised Figure 2—figure supplement 1A). We have added this information to the revised manuscript:

Line: 198-202

“To corroborate the distribution of GFP-Minar2, we generated a FLAG-tagged Minar2 construct, *myo6b*:GFP-P2A-FLAG-Minar2, which allowed labeling of hair cells by GFP and localization of Minar2 by the small FLAG tag. The results showed that FLAG-Minar2 was similarly localized to the stereocilia and the apical region of hair cells (Figure 2—figure supplement 1A).”

2. As stated above, can the authors perform EM? Also, are there defects in the neuromasts in mutants?

We are not equipped to perform electron microscopy. In the neuromasts, our data showed that the kinocilia were disorganized in the *minar2* mutant larvae, in contrast to the normal bundled-together morphology in the wild-type animals (revised Figure 2—figure supplement 1D). We think that this defect in the morphology of kinocilia bundles is caused by a certain abnormality in hair cells since a disorganized kinocilia bundle was also observed in neuromasts following mechanical damage (Figure 9 in Holmgren et al. 2021). We have added this information to the revised manuscript:

Line: 229-235

“We also found that the kinocilia of lateral line hair cells were disorganized in the *minar2* mutant larvae, in contrast to the normal bundled together morphology in the wild-type animals (Figure 2—figure supplement 1D, for 5 dpf, *minar2^fs139^*: p<0.001, *minar2^fs140^*: p<0.01; for 8 dpf, *minar2^fs139^*: p<0.001, *minar2^fs140^*: p<0.01. Fisher’s exact test). The disorganized kinocilia bundle was a reminiscence of the kinocilia morphology seen in neuromasts following mechanical injury (Holmgren et al. 2021).”

3. Figure 1A do prim 1- and prim 2-derived neuromasts express minar2? Do anterior neuromasts express minar2?

Our in situ hybridization results showed that *minar2* was expressed in all neuromasts, including prim 1 and 2 derived neuromasts and anterior lateral line (aLL) neuromasts. We have added a representative figure showing *minar2* expression in aLL neuromasts (revised Figure 1—figure supplement 1A).

4. The author should figure out a way to determine if the PM of stereocilia has higher levels of cholesterol than that of the basolateral PM. They need to take into account that there is a much higher level of PM in stereocilia.

We thank the reviewer for raising this concern. We see that the plasma membrane is highly folded in the stereocilium and there is likely much more plasma membrane in any given area of the stereocilium than that of the basolateral membrane.

To address this question, in the revised manuscript, we quantified and compared the distribution of PM-GFP with that of D4H-mCherry in individual inner ear hair cells. The general plasma membrane probe PM-GFP was expected to indiscriminately label the plasma membrane in both the stereocilium and the basolateral cell body, so measurement of PM-GFP signal in the stereocilium and the basolateral cell body may provide an estimate of the relative amount of the plasma membranes in the stereocilium and the basolateral membranes. As expected, although the mean area of the stereocilium region was only 16.5% of that of basolateral regions, the mean PM-GFP fluorescence within the stereocilium regions was 73.7% of that of the basolateral regions. This over-representation of PM-GFP in the stereocilium region is likely because that the plasma membrane in the stereocilium region is folded multiple times to give rise to a denser plasma membrane in the stereocilia (about 4.5-times denser by simple division).

Strikingly, the mean D4H-mCherry fluorescence within the stereocilium regions was 567.7% of that of the basolateral regions (enriched 34.4-fold by simple division), which was 7.7 times higher than that of PM-GFP (567.7% versus 73.7%, Figure 4—figure supplement 1B). Thus the disproportional distribution of D4H-mCherry in the stereocilium is much larger than what is expected from the distribution of the general plasma membrane probe PM-GFP. These results supported that there was an enrichment of D4H-mCherry in the stereocilium membrane over the basolateral membranes. We have added this information to the revised manuscript.

Line: 349-364

“We quantified and compared the distribution of PM-GFP with that of D4H-mCherry in individual inner ear hair cells. We segmented the stereocilium and basolateral cell body region of the hair cells, and then measured the areas and the fluorescence values within the segmented regions. The results showed that although the mean area of the stereocilium region was only 16.5±0.1% of that of basolateral regions, the mean PM-GFP fluorescence within the stereocilium regions was 73.7±4.0% of that of the basolateral regions. The over-representation of PM-GFP signal in the stereocilium region is likely because the plasma membrane in the stereocilium region is folded multiple times to give rise to a denser plasma membrane in the stereocilium region. Strikingly, the mean D4H-mCherry fluorescence within the stereocilium regions was 567.7±80.1% of that of the basolateral regions, which was 7.7 times higher than that of PM-GFP (p < 0.0001, Mann-Whitney test, Figure 4—figure supplement 1B). Therefore, the disproportional distribution of D4H-mCherry in the stereocilium is much larger than what is expected from the distribution of the general plasma membrane probe PM-GFP, indicating that there was an enrichment of D4H-mCherry in the stereocilium membrane over the basolateral membranes.”

5. Perform a genomics search and let us know if there are 2 copies of Kiaa1024L/Minar2 in the zebrafish genome.

We performed BLASTP and TBLASN search of the GRCz11 zebrafish genome build and we found a single *kiaa1024l/minar2* gene. There are two *kiaa1024/minar1/ubtor* gene orthologs (named *ubtora* and *ubtorb* in (Zhang et al., 2018)), located on chromosomes 7 and 18, respectively. The single *kiaa1024l/minar2* gene ortholog is located on chromosome 10. We have added this information to the revised manuscript.

Line: 96-99

“Based on genome annotations and BLAST search results, there are two *kiaa1024/minar1/ubtor* gene orthologs (named *ubtora* and *ubtorb* in (Zhang et al., 2018)), and a single *kiaa1024l/minar2* gene ortholog in the zebrafish genome.”

References

El-Brolosy, M. A., Kontarakis, Z., Rossi, A., Kuenne, C., Gunther, S., Fukuda, N., Kikhi, K., Boezio, G. L. M., Takacs, C. M., Lai, S. L., Fukuda, R., Gerri, C., Giraldez, A. J., and Stainier, D. Y. R. (2019). Genetic compensation triggered by mutant mRNA degradation. Nature, 568(7751), 193-197. https://doi.org/10.1038/s41586-019-1064-z

Farrand, A. J., LaChapelle, S., Hotze, E. M., Johnson, A. E., and Tweten, R. K. (2010). Only two amino acids are essential for cytolytic toxin recognition of cholesterol at the membrane surface. Proc Natl Acad Sci U S A, 107(9), 4341-4346. https://doi.org/10.1073/pnas.0911581107

Linkert, M., Rueden, C. T., Allan, C., Burel, J. M., Moore, W., Patterson, A., Loranger, B., Moore, J., Neves, C., Macdonald, D., Tarkowska, A., Sticco, C., Hill, E., Rossner, M., Eliceiri, K. W., and Swedlow, J. R. (2010). Metadata matters: access to image data in the real world. J Cell Biol, 189(5), 777-782. https://doi.org/10.1083/jcb.201004104

Ma, Z., Zhu, P., Shi, H., Guo, L., Zhang, Q., Chen, Y., Chen, S., Zhang, Z., Peng, J., and Chen, J. (2019). PTC-bearing mRNA elicits a genetic compensation response via Upf3a and COMPASS components. Nature, 568(7751), 259-263. https://doi.org/10.1038/s41586-019-1057-y

Zhang, H., Zhang, Q., Gao, G., Wang, X., Wang, T., Kong, Z., Wang, G., Zhang, C., Wang, Y., and Peng, G. (2018). UBTOR/KIAA1024 regulates neurite outgrowth and neoplasia through mTOR signaling. PLoS Genet, 14(8), e1007583. https://doi.org/10.1371/journal.pgen.1007583